# MMaDA-Parallel: Multimodal Large Diffusion Language Models for Thinking-Aware Editing and Generation

**Ye Tian**[1,2*]    **Ling Yang**[3*]  **Jiongfan Yang**[1]   **Anran Wang**[2]   **Yu Tian**[2]   **Jiani Zheng**[2]
**Haochen Wang**[2,4]   **Zhiyang Teng**[2]   **Zhuochen Wang**[2]   **Yinjie Wang**[5]
**Yunhai Tong**[1†]   **Mengdi Wang**[3†]   **Xiangtai Li**[2†]
[1]Peking University [2]ByteDance [3]Princeton University [4]CASIA [5]The University of Chicago
Huggingface: MMaDA-Parallel-Model    Code: MMaDA-Parallel-Code

## Abstract

While thinking-aware generation aims to improve performance on complex tasks, we identify a critical failure mode where existing sequential, autoregressive approaches can paradoxically degrade performance due to error propagation. To systematically analyze this issue, we propose ParaBench, a new benchmark designed to evaluate both text and image output modalities. Our analysis using ParaBench reveals that this performance degradation is strongly correlated with poor alignment between the generated reasoning and the final image. To resolve this, we propose a parallel multimodal diffusion framework, **MMaDA-Parallel**, that enables continuous, bidirectional interaction between text and images throughout the entire denoising trajectory. MMaDA-Parallel is trained with supervised fine-tuning and then further optimized by Parallel Reinforcement Learning (ParaRL), a novel strategy that applies semantic rewards along the trajectory to enforce cross-modal consistency. Experiments validate that our model significantly improves cross-modal alignment and semantic consistency, achieving a 6.9% improvement in Output Alignment on ParaBench compared to the state-of-the-art model, Bagel, establishing a more robust paradigm for thinking-aware image synthesis.

## 1 Introduction

Recent advances in multimodal generative models have achieved remarkable progress in instruction-based image generation and editing (Esser et al., 2024a; Labs, 2024; Wei et al., 2024; Liu et al., 2025b). Given diverse textual prompts, these models can produce visually coherent and semantically aligned results across a wide range of tasks. However, these models often struggle with **complex instructions that require reasoning over world knowledge**, frequently leading to incorrect editing and generation (Wu et al., 2025c; Niu et al., 2025; Zhao et al., 2025). To mitigate this gap, recent studies have introduced intermediate reasoning steps before visual generation (Fang et al., 2025; Jiang et al., 2025a; Deng et al., 2025a). In these approaches, textual reasoning is first performed to guide subsequent image editing and generation. Such explicit reasoning has proven effective in improving the quality and consistency of image editing and generation (Deng et al., 2025a).

Despite the general effectiveness of incorporating a reasoning process prior to image synthesis, we observe a counterintuitive and critical phenomenon. On certain benchmarks (Wu et al., 2025c), the inclusion of reasoning can in fact **reduce the semantic fidelity of the generated images**. For example, in Figure 1(a), a "thinking-aware" model starts with correct reasoning but then shifts to refining minor details like background textures. This reduces attention on the primary subject and causes the final edit to misidentify it completely. The resulting image thus deviates from the user's core instruction and even contradicts its own thinking prompt, leading to a clear performance drop. This raises a crucial question: *What underlies this performance degradation?*

---

*Equal Contribution.
†Correponding Authors

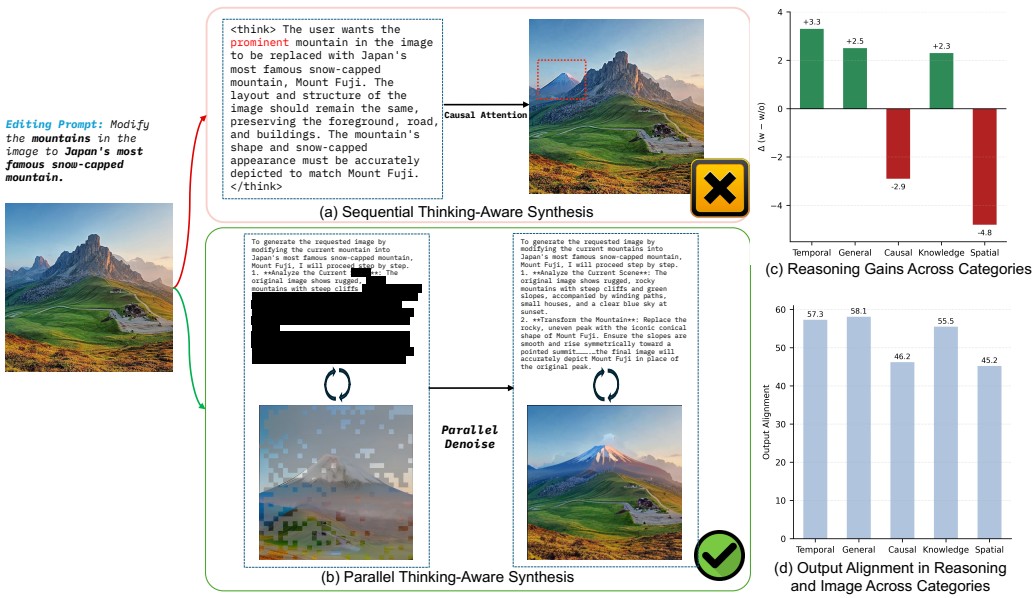

Figure 1: Sequential vs. parallel thinking-aware image synthesis. (a) Sequential generation (**Bagel, GPT4o**) may suffer from vague or incorrect reasoning. (b) Parallel generation aligns text and image at each denoising step, reducing hallucination and errors. (c) Quantitative comparison shows reasoning can degrade performance in certain categories. (d) Poorer categories also exhibit weaker reasoning–image alignment, highlighting the need for stronger cross-modal alignment.

Based on these failure cases, we hypothesize that the degradation stems from the reasoning text itself. However, this hypothesis is difficult to verify with existing benchmarks (Wu et al., 2025c; Zhao et al., 2025). These benchmarks only evaluate the final image against the initial prompt, but cannot evaluate the intermediate reasoning step or its alignment with the final output.

Therefore, we introduce *ParaBench*, our new benchmark designed to explicitly evaluate this output alignment between a model's generated reasoning and its final image. Using ParaBench to evaluate the state-of-the-art model Bagel (Deng et al., 2025a), we find a strong correlation: performance degradation occurs precisely in categories where output alignment is weakest (Figure 1(d)). We attribute this to the compounding errors inherent in sequential autoregressive models, where ambiguous or incomplete reasoning provides unreliable guidance for the subsequent image generation, ultimately degrading the final output.

Thus, while pre-reasoning can in principle enhance multimodal generation, its reliance on an autoregressive pipeline makes the process vulnerable to error accumulation and semantic drift. Recently, another line of work has explored discrete diffusion models for text or image generation (Nie et al., 2025; Yang et al., 2025a; Ye et al., 2025a), which remove the token-by-token constraint of autoregression and instead employ confidence-based sampling to achieve greater global consistency. Inspired by these advances, we ask: **What if multimodal models could generate text and images in parallel?** Such a paradigm directly addresses the limitations of AR reasoning: text and images can attend to each other at every denoising step, avoiding the propagation of hallucinations and vague priors while grounding textual descriptions in visual evidence.

Building on this insight, we propose a purely diffusion-based framework for *parallel text–image generation*, where cross-modal interaction is maintained throughout the trajectory to ensure robust and semantically faithful multimodal editing and generation, as shown in Figure 1(b)).

To train this framework, we first establish a thinking-aware data curation pipeline. We prompt a powerful VLM with data triplets (⟨input image, edit instruction, output image⟩) sourced from widely-adopted image editing and generation datasets. The VLM is tasked to generate a reasoning trace that explains the edit process. This pipeline yields a training dataset of quadruplets: ⟨input image, instruction, reasoning trace, output image⟩, designed to elicit the models' reasoning

and generation capabilities. We use this dataset to perform supervised fine-tuning on MMaDA (Yang et al., 2025a). This parallel version, **MMaDA-Parallel**, demonstrates higher output consistency compared to sequential baselines.

Notably, such consistency is observed not only in the final outputs but also **throughout the generation trajectory.** We observe that during the parallel denoising process, the image region corresponding to a specific semantic concept is often refined simultaneously with its textual counterpart. However, standard SFT and conventional reinforcement learning algorithms optimize for the final outcome only. This output-level supervision is too coarse to enforce the fine-grained, stepwise alignment we observe and cannot guarantee consistency at intermediate steps. To fully leverage this trajectory-level consistency, we draw inspiration from process-level and trajectory-level optimization methods (Li & Li, 2024; Wang et al., 2025) and introduce *Parallel Reinforcement Learning (ParaRL)*. Instead of focusing solely on the final outcome, ParaRL incorporates stepwise semantic supervision to refine alignment along the denoising trajectory. Our experiments demonstrate that this trajectory-level optimization provides a more granular and effective signal for diffusion models compared to traditional output-level supervision.

Extensive quantitative and qualitative results validate the effectiveness of MMaDA-Parallel for thinking-aware image editing and generation, and further highlight the additional gains achieved through ParaRL. Our contributions can be summarized as follows:

1. **In-depth Benchmarking and Analysis of Thinking-aware Image Synthesis.** We propose ParaBench, which systematically evaluates thinking-aware image generation and editing, focusing on text and image quality and their alignment.

2. **Parallel Multimodal Diffusion Framework.** We propose a purely discrete diffusion-based approach for parallel thinking-aware image editing and generation, which enables bidirectional attention between modalities at every denoising step and effectively alleviates the error accumulation of autoregressive pipelines.

3. **Parallel Reinforcement Learning.** We introduce a parallel reinforcement learning strategy, *ParaRL*, which assigns semantic rewards along the denoising trajectory, further enhancing alignment between the output modalities and the overall performance.

4. **Extensive Evaluation and State-of-the-Art Alignment.** Our comprehensive experiments validate the framework, establishing state-of-the-art performance among open-source models with a 6.9% gain in Output Alignment over Bagel on our ParaBench benchmark, while maintaining comparable performance on single-modality metrics.

## 2 RELATED WORK

Recent progress in multimodal models for image understanding, generation, and editing has been rapid, yet most approaches remain constrained to single-modal generation conditioned on multiple modalities (Esser et al., 2024b; Wu et al., 2025a; Labs et al., 2025; Bai et al., 2025). To improve the accuracy and fidelity of multimodal generation, a growing line of work has explored introducing a textual *Chain-of-Thought* reasoning process before image generation or editing. We refer to this paradigm as **thinking-aware image generation and editing**. For instance, early efforts such as Chameleon (Team, 2024) and Mogao (Liao et al., 2025) investigated interleaved generation, enabling interleaving sequences of text and image tokens. Image-CoT (Guo et al., 2025b) and GoT (Fang et al., 2025) incorporated CoT reasoning before image synthesis, demonstrating that reasoning traces can enhance generation quality. Bagel (Deng et al., 2025a) further extended this idea by integrating chain-of-thought reasoning into both image generation and editing, enabling more flexible and semantically aligned outputs. Building on this direction, follow-up works such as OmniGen2 (Wu et al., 2025b) and IRG (Huang et al., 2025a) introduced reflective reasoning after image generation, using multi-turn textual feedback to refine visual outputs iteratively. Most existing methods, however, rely on a sequential autoregressive interleaved pipeline, which could limit direct cross-modal interaction and make the model prone to error accumulation from imperfect reasoning traces. Exploring a parallel generation framework that enables more interaction within output modalities is still lacking in this scenario. (More related work can be found in Appendix C).

Table 1: **Thinking may degrade the performace of visual synthesis.** Bagels' performance comparison on ParaBench editing tasks with and without thinking. We also report the reasoning quality (Text Qual.) and cross-modal alignment (Output Align.).

| Editing Category | w/o Thinking | w/ Thinking | Δ (w/ − w/o) | Text Qual. ↑ | Output Align.↑ |
|---|---|---|---|---|---|
| Temporal | 72.3 | 75.6 | +3.3 | 92.6 | 57.3 |
| General | 68.9 | 71.4 | +2.5 | 86.2 | 58.1 |
| Causal | 70.1 | 67.2 | **−2.9** | **75.3** | **46.2** |
| Knowledge | 74.5 | 76.8 | +2.3 | 87.8 | 55.5 |
| Spatial | 69.8 | 65.0 | **−4.8** | **73.2** | **45.2** |

## 3 MMADA-PARALLEL

### 3.1 FINDINGS AND BENCHMARKING ON THINKING-AWARE SYNTHESIS

To investigate whether pre-generation reasoning genuinely enhances performance, we conduct a controlled study on image editing tasks, which provides a clearer instruction-grounded evaluation than naive synthesis. We sample inputs from established benchmarks (Wu et al., 2025c; Zhao et al., 2025) and generate paired outputs using Bagel (Deng et al., 2025a)—an advanced, open-source, unified model supporting thinking-aware generation—with and without thinking. We report the average editing evaluation metrics in Kris-Bench (Wu et al., 2025c) in Figure 1(c) and also Table 1.

**Findings.** While the reasoning step enhanced performance on most tasks, a notable countertrend emerged: performance declined in a significant subset of cases, about 23%, particularly in complex compositional edits. A closer analysis reveals that these failures often stemmed from low-quality or vague reasoning text, which misguides the image generation process. This exposes a critical gap in existing protocols: they evaluate the final image but ignore the quality of the intermediate reasoning—the other generated modality.

**Benchmarking mixed modalities.** This analysis reveals a fundamental limitation in current evaluation paradigms: existing benchmarks (Wu et al., 2025c; Zhao et al., 2025; Ghosh et al., 2023) only evaluate images, ignoring the quality of the reasoning itself and its consistency with the image. To address this gap, we introduce **ParaBench**, a new benchmark specifically designed for the comprehensive evaluation of thinking-aware image synthesis. ParaBench comprises 300 challenging prompts, split into 200 for editing and 100 for generation. The editing prompts are meticulously curated to test a wide spectrum of abilities, covering not only general operations (e.g., add, remove, replace) but also complex tasks requiring reasoning. The 100 generation prompts focus on open-ended creative synthesis of complex scenes. We evaluate models on ParaBench using the GPT-4.1 across six fine-grained aspects: for the textual output, we assess Text Quality and Text Alignment; for the visual output, we evaluate Image Quality, Image Alignment, and Image Consistency; and finally, the overall Output Alignment between them. More details are included in Appendix G.

To demonstrate ParaBench's diagnostic capabilities, we apply it to a representative baseline, Bagel. While full quantitative results are presented in Sec A, Table 1 highlights a crucial finding by focusing on two key metrics: **Text Quality** and **Output Alignment**. The results reveal a clear correlation between the quality of the reasoning step and the final performance. Notably, the categories that exhibited performance degradation also suffered from significant drops in both reasoning quality and reasoning-image synergy. This pattern strongly suggests that poor reasoning does not merely fail to provide helpful guidance but actively misleads the generation process, validating the necessity of explicitly improving the synergy between text and image generation.

**Motivations on parallel multimodal diffusion.** Our benchmarking results reveal a critical limitation in current thinking-aware generation: *the sequential generation paradigm*, where reasoning precedes image synthesis, creates a rigid dependency that can propagate errors and limit cross-modal synergy. When reasoning quality degrades, it directly undermines the subsequent image generation, as demonstrated by the correlated performance drops in spatial and temporal editing tasks. To address this fundamental issue, we propose a parallel unified multimodal diffusion framework that enables simultaneous generation of both reasoning text and images, fostering genuine multimodal collaboration while eliminating the error propagation inherent in sequential approaches.

## 3.2 Basic Algorithm and Architecture

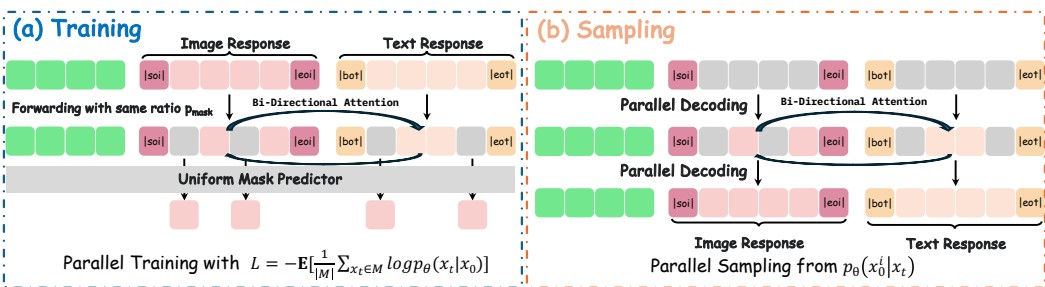

Figure 2: Parallel Generation Architecture: During (a) training, image and text responses are masked and predicted in parallel with a uniform mask predictor, optimized by the masked token likelihood objective. During (b) sampling, the model performs parallel decoding to generate both image and text responses jointly, enabling efficient multimodal response generation.

Discrete diffusion models have demonstrated strong performance for both image and text generation (Bai et al., 2024; Nie et al., 2025; Zhu et al., 2025). Building on the unified discrete-diffusion view, MMaDA (Yang et al., 2025a) demonstrates that a single diffusion framework can jointly model multiple modalities; however, its decoding remains *sequential* across modalities. To overcome this limitation, we propose a *parallel* multimodal diffusion framework that: (i) represents all modalities as discrete tokens, (ii) arranges them in an interleaved sequence with bidirectional attention, and (iii) employs a single mask predictor shared across modalities, enabling synchronous denoising for both text and images. An overview of this framework is shown in Figure 2.

**Interleaved discrete sequence layout.** Following the MMaDA framework (Yang et al., 2025a), we process both text and images within a unified discrete token space. Specifically, we tokenize text using the LLaDA tokenizer (Nie et al., 2025) and encode images into a grid of discrete visual tokens using a pretrained MAGVIT-v2 (Yu et al., 2023) quantizer. These tokenized modalities are then serialized into a single interleaved sequence, using explicit sentinels and task tags to enable full bidirectional cross-modal attention:

```
Input:  <|task|><|soi|>[img]<|eoi|><|bos|>[text]<|eos|>
Output: <|soi|>[output img]<|eoi|><|bos|>[output text]<|eos|>
```

During training, we concatenate the input and output templates into a single sequence, allowing the model to attend from outputs to inputs within a unified context. The task token `<|task|>` is instantiated differently depending on the scenario, with `<|thinkgen|>` used for thinking-aware generation and `<|thinkedit|>` used for thinking-aware editing. This single-sequence design eliminates the ordering asymmetry and exposure bias introduced by autoregressive cross-modal pipelines.

**Training objective.** Let $x_0 \in \{1, \ldots, V\}^L$ denote the concatenated training sequence (input part followed by output part), where $L$ is the total number of tokens in the sequence. We keep the input part static and apply noise only to the output part. At a sampled timestep $t \in \{1, \ldots, T\}$, for each token in the *output* part we replace it with `[MASK]` with probability $\beta_t$ and keep it unchanged with probability $1 - \beta_t$; tokens in the *input* part are left unchanged:

$$x_t^{(i)} = \begin{cases} x_0^{(i)} & \text{if } i \text{ in input,} \\ x_0^{(i)} \text{ with prob. } (1 - \beta_t), \text{ [MASK] with prob. } \beta_t & \text{if } i \text{ in output.} \end{cases} \quad (1)$$

Equivalently, for positions in the output, the absorbing-state marginal after $t$ steps is $q(x_t \mid x_0) = \alpha_t x_0 + (1 - \alpha_t) \mathbf{m}$ where $\alpha_t = \prod_{k=1}^{t}(1 - \beta_k)$, and $\mathbf{m}$ is the one-hot distribution of `[MASK]`.

The parallel diffusion model $p_\theta(\cdot \mid x_t)$ is formulated as a unified masked-token predictor over the joint vocabulary of text and image tokens. Let $i \in 1, \ldots, L$ denote token positions in the

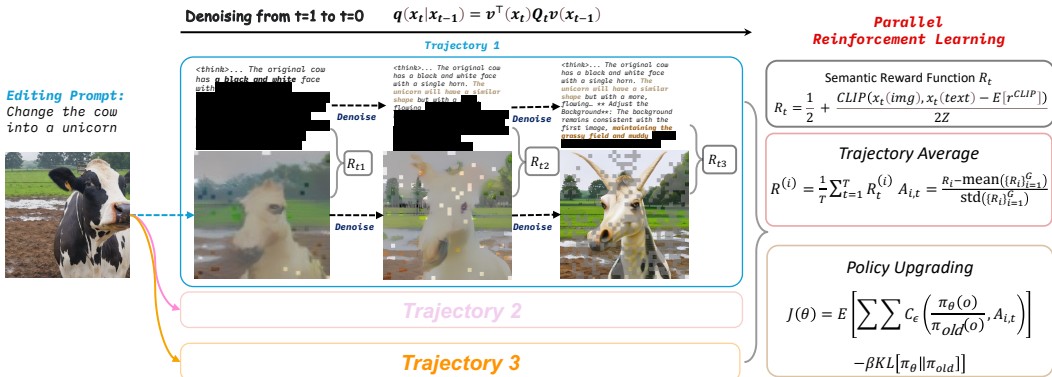

Figure 3: Overview of our proposed Parallel Reinforcement Learning (ParaRL). Rather than optimization only to the final denoised outputs, ParaRL introduces reward signals along the entire denoising trajectory, reinforcing semantic alignment consistently throughout the generation process.

concatenated input–output sequence. Since only the output segment is noised during diffusion, the model predicts ground-truth tokens $x_0$ at the currently masked positions within this segment. To better balance the training dynamics across modalities, we make the timestep-dependent loss weight modality-specific: tokens in the *output image* segment and the *output text* segment are assigned separate weights, $w_{\text{img}}(t)$ and $w_{\text{text}}(t)$. For compactness, we write the objective using a unified token-aware weight function $w(t, i)$. We optimize a timestep-reweighted cross-entropy:

$$\mathcal{L}_{\text{parallel}}(\theta) = -\mathbb{E}_{t, x_0, x_t} \left[ \sum_{i=1}^{L} w(t, i) \, \mathbf{1}\big[x_t^{(i)} = \texttt{[MASK]}\big] \log p_\theta\big(x_0^{(i)} \mid x_t\big) \right], \qquad (2)$$

where $\mathbf{1}[\cdot]$ is the indicator function and

$$w(t, i) = \begin{cases} w_{\text{img}}(t), & \text{if } i \text{ lies in the } \textit{output image} \text{ segment,} \\ w_{\text{text}}(t), & \text{if } i \text{ lies in the } \textit{output text} \text{ segment.} \end{cases}$$

We empirically find that applying a timestep-dependent weighting $w_{\text{text}}(t) = 1/t$ for text tokens and a constant weighting $w_{\text{img}}(t) = 1$ for image tokens substantially stabilizes the training of image quality and output alignment. We illustrate this process in Figure 2(a) and include detailed additional preliminaries with ablations in Appendix D.

**Parallel denoising with dual schedulers.** Decoding proceeds along a shared diffusion time axis $t_T \rightarrow \cdots \rightarrow t_0$, as is shown in Figure 2(b). We define two modality-specific schedulers, $u_{\text{img}}(t), u_{\text{text}}(t) \in [0, 1]$, which specify the target proportion of unmasked tokens at step $t$. At each reverse step: (i) the model jointly predicts distributions for all currently masked positions; (ii) for each modality, a fraction of tokens is sampled (e.g., via confidence-based sampling), while the remaining positions are retained as $\texttt{[MASK]}$. Because attention is bidirectional across the *entire* sequence, text and image can inform each other at every step of decoding. In our experiments, the text schedule is implemented as a fully linear reveal schedule combined with semi-autoregressive confidence-based decoding Nie et al. (2025), while the image schedule follows a cosine reveal schedule with global confidence-based decoding. More details can be found in Appendix E.

### 3.3 POST TRAINING WITH PARALLEL REINFORCEMENT LEARNING

**Supervised Finetuning for Parallel Synthesis** A key challenge in our approach is that existing generation and editing datasets lack the reasoning traces required for our parallel synthesis framework. To address this, we construct a suitable training dataset by first aggregating samples from various sources. For each sample comprising an input image (for editing tasks), an instruction, and the final output image, we employ a multimodal LLM (Qwen-2.5-VL in our implementation) to generate a corresponding reasoning trace. Further details on the dataset construction process, including the sources and categories, are provided in Appendix F. We then use this dataset to perform

supervised fine-tuning on MMaDA (Yang et al., 2025a). This process adapts it into a parallel variant capable of performing thinking-aware synthesis, where reasoning and generation occur concurrently.

**Synergy along the denoising trajectory.** While analyzing generations from the finetuned model, we observe that certain semantic concepts emerge *synchronously* in text and image at intermediate denoising steps. As illustrated in Figure 4, when tasked to change a shirt to a "vibrant rainbow color," the specific color words and their corresponding visual features appear at the same timestep. This observation leads to a key insight: cross-modal alignment is not an endpoint phenomenon but is progressively established **throughout the generation trajectory**. This implies that supervision applied to these intermediate steps, not just the final output, can further improve this alignment.

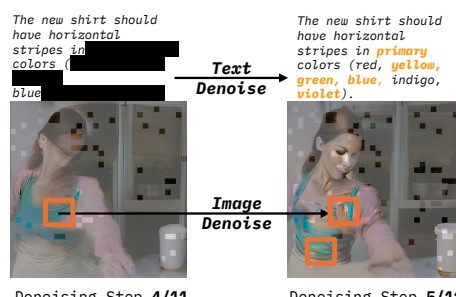

Figure 4: Synergy of sampling. Given the prompt: "change the blue shirt to a vibrant rainbow color," the specific color decoding in text and image emerges at the same step.

**Parallel reinforcement learning with trajectory optimization.** Building on this insight, we further introduce Parallel Reinforcement Learning (ParaRL), a novel training paradigm that directly leverages this intermediate cross-modal synergy. Instead of rewarding only the final output, ParaRL uses the alignment between text and image tokens at each denoising step as a dense reward signal.

Specifically, for a given query $Q$, the generated response is a full trajectory $\tau_i \triangleq \big(\tau_i(1), \ldots, \tau_i(|\tau_i|)\big)$, where $|\tau_i|$ is the total number of denoising steps and $\tau_i(t)$ is the set of tokens decoded at step $t$. While this formulation provides a step-wise reward $r_{i,t}$ for each intermediate response $\tau_i(t)$, optimizing over the entire dense trajectory is computationally prohibitive. To make training feasible, we adopt a sparse optimization strategy. During each online rollout, we pre-select sampling steps $s$ and fix subset of step indices $S \subset \{1, \ldots, |\tau_i|\}, |S| = s$ and only compute rewards $r_{i,t}$ and their corresponding standardized advantages $A_{i,t}$ for timesteps $t \in S$. We adapt a diffusion GRPO objective (Gong et al., 2025) that accommodates token-level likelihood ratios with advantages calculated at these sampled steps:

$$
\mathcal{J}_{\text{policy}}(\theta) = \mathbb{E}_{\substack{Q \sim D_{\text{task}} \\ \{\tau_i\}_{i=1}^G \sim \pi_{\text{old}}(\cdot|Q)}} \left[ \sum_{i=1}^G \sum_{t \in S} \frac{1}{|\tau_i(t)|} \sum_{o \in \tau_i(t)} C_\epsilon \left( \frac{\pi_\theta(o \mid Q, \tau_i(1{:}t-1))}{\pi_{\text{old}}(o \mid Q, \tau_i(1{:}t-1))}, A_{i,t} \right) \right] \\
- \beta \, \text{KL}\big[\pi_\theta \,\|\, \pi_{\text{old}}\big],
\tag{3}
$$

where $C_\epsilon(r, A) \triangleq \min\big(rA, \, \text{clip}(r, \, 1-\epsilon, \, 1+\epsilon)\, A\big)$. In this objective, the summation is performed over the sparsely sampled steps $t \in S$. The term $o$ ranges over all tokens within the state $\tau_i(t)$ at a sampled step $t$, and $\tau_i(1{:}t-1)$ denotes the full history of tokens generated prior to step $t$. Finally, $\pi_{\text{old}}$ is the behavior policy for generating rollouts, and $\beta$ controls the KL penalty strength.

**Trajectory reward design.** In typical trajectory-level optimization frameworks, a well-trained process reward model (PRM) (Li & Li, 2024) or value function Wang et al. (2025) is often required, since intermediate partial outputs usually lack sufficient semantic information for reliable evaluation. Surprisingly, in our parallel text–image generation setting, we find that intermediate fragments are already semantically meaningful. For instance, even partially decoded text tokens often reveal enough semantic cues to compute alignment with the simultaneously generated image content, as illustrated in Figure 3. This observation allows us to bypass the need for a dedicated PRM: we directly employ *semantic alignment* between text and image as the reward signal.

Unlike tasks with binary rewards (e.g., mathematical reasoning), our cross-modal alignment objective provides a continuous reward signal. However, the naive CLIP score, which serves as our reward source, can exhibit high variance and an arbitrary scale, making it unstable for direct use in reinforcement learning. To ensure training stability, we therefore apply a normalization scheme

Table 2: **Main results on *ParaBench***. Evaluation across editing and generation tasks. For non-thinking image editing or generation models, text evaluation and output alignment are unavailable.

| Model | Text Qual. | Text Align. | Image Cons. | Image Align. | Image Qual. | Output Align. | Overall |
|---|---|---|---|---|---|---|---|
| **Open-source models (Non-thinking)** | | | | | | | |
| Flux.1-Dev | - | - | - | 65.2 | 77.5 | - | - |
| Qwen-Image | - | - | - | 67.2 | **84.2** | - | - |
| Flux.1-Kontext | - | - | 77.9 | 65 | 84 | - | - |
| Qwen-Image-Edit | - | - | **78.2** | **73.5** | 84.1 | - | - |
| Bagel (w/o think) | - | - | 72.2 | 50.3 | 80.1 | - | - |
| **Closed-source models** | | | | | | | |
| GPT-4o | 92.5 | 93.4 | 86.2 | **85.7** | 88.1 | **69.5** | **85.9** |
| Gemini-2.5 | **94.1** | **95.2** | **88.5** | 76.2 | **90.2** | 63.4 | 84.6 |
| **Open-source models (Thinking-aware)** | | | | | | | |
| Bagel (w/ think) | **82** | 70.5 | **76.7** | **63.4** | **81.5** | 52.9 | 71.2 |
| Show-o* (tuned) | 75.2 | 70.7 | 69.1 | 57.5 | 78.5 | 48.9 | 66.6 |
| **MMaDA-Parallel w/o ParaRL** | 76.5 | 70.4 | 70.5 | 58.2 | 80.5 | 51.5 | 67.9 |
| **MMaDA-Parallel w/ ParaRL** | 80.4 | **71** | 73.4 | 63.2 | 81.2 | **59.8** | **71.5** |

inspired by prior work in RL with continuous rewards (Liu et al., 2025a). We begin by estimating the mean $\mu_{\text{CLIP}}$ and standard deviation $\sigma_{\text{CLIP}}$ of CLIP scores across the training distribution, where we compute on a random 1% subset of the data. Let $c_{i,t} = R^{\text{CLIP}}(\text{text}(\tau_i(t)), \text{image}(\tau_i(t)))$ be the raw CLIP score for the content generated at step $t$. We first standardize this score to obtain $\hat{c}_{i,t}$ using $\hat{c}_{i,t} = \frac{c_{i,t} - \mu_{\text{CLIP}}}{\sigma_{\text{CLIP}}}$. This standardized score is then clipped to the range $[-1, 1]$ and linearly rescaled to yield the final reward $R_{i,t}$, which is bounded within $[0, 1]$:

$$R_{i,t} = \frac{1}{2}\left(1 + \text{clip}(\hat{c}_{i,t}, -1, 1)\right) \tag{4}$$

The corresponding advantages $A_{i,k}$ used in Eq. 3 are then obtained by standardization over the rollouts: $A_{i,t} = \frac{R_{i,t} - \text{mean}\left(\{R_{j,t}\}_{j=1}^{G}\right)}{\text{std}\left(\{R_{j,t}\}_{j=1}^{G}\right)}$

## 4 EXPERIMENTS

### 4.1 IMPLEMENTATION DETAILS

**Training and datasets.** Our final model, MMaDA-Parallel, is trained in a two-stage process. We begin with supervised finetuning (SFT) on the MMaDA-MixCoT model, which integrates a LLaDA-8B text backbone with a MagVIT-v2 image tokenizer. For this stage, we construct a new dataset of 150K thinking-aware image editing and generation pairs, meticulously sourced and filtered from multiple existing benchmarks. In the second stage, we apply reinforcement learning with a GRPO-based objective. To enhance training efficiency, this RL stage focuses on the most challenging 10% of the SFT examples, optimizing the policy online to improve cross-modal semantic alignment. More details of the dataset and training details can be found in Appendix F and H.

**Evaluation setup.** We conduct our primary evaluation on the ParaBench benchmark, which was introduced in the Method section. We employ an LLM-as-a-judge framework (GPT-4.1) to assess performance across the six fine-grained metrics previously described, covering text quality, image fidelity, and cross-modal alignment. The prompts used for the LLM judge are detailed in the Appendix G. Our **MMaDA-Parallel** is compared against state-of-the-art thinking-aware models, including Bagel (Deng et al., 2025a), GPT-4o, and Gemini-2.5, as well as leading image-only generators like Qwen-Image (Wu et al., 2025a), Qwen-Image-Edit (Wu et al., 2025a), Flux.1-dev (Labs, 2024) and Flux.1-Kontext (Labs et al., 2025).

### 4.2 MAIN RESULTS

Table 2 reports the overall performance on our ParaBench benchmark. Our proposed method, **MMaDA-Parallel**, achieves the highest *Output Alignment* among all open-source models, confirming the effectiveness of its parallel multimodal decoding and trajectory-level optimization. In terms of general text and image quality, **MMaDA-Parallel** performs on par with Bagel, despite Bagel being trained on a dataset nearly three orders of magnitude larger. Compared to leading closed-source

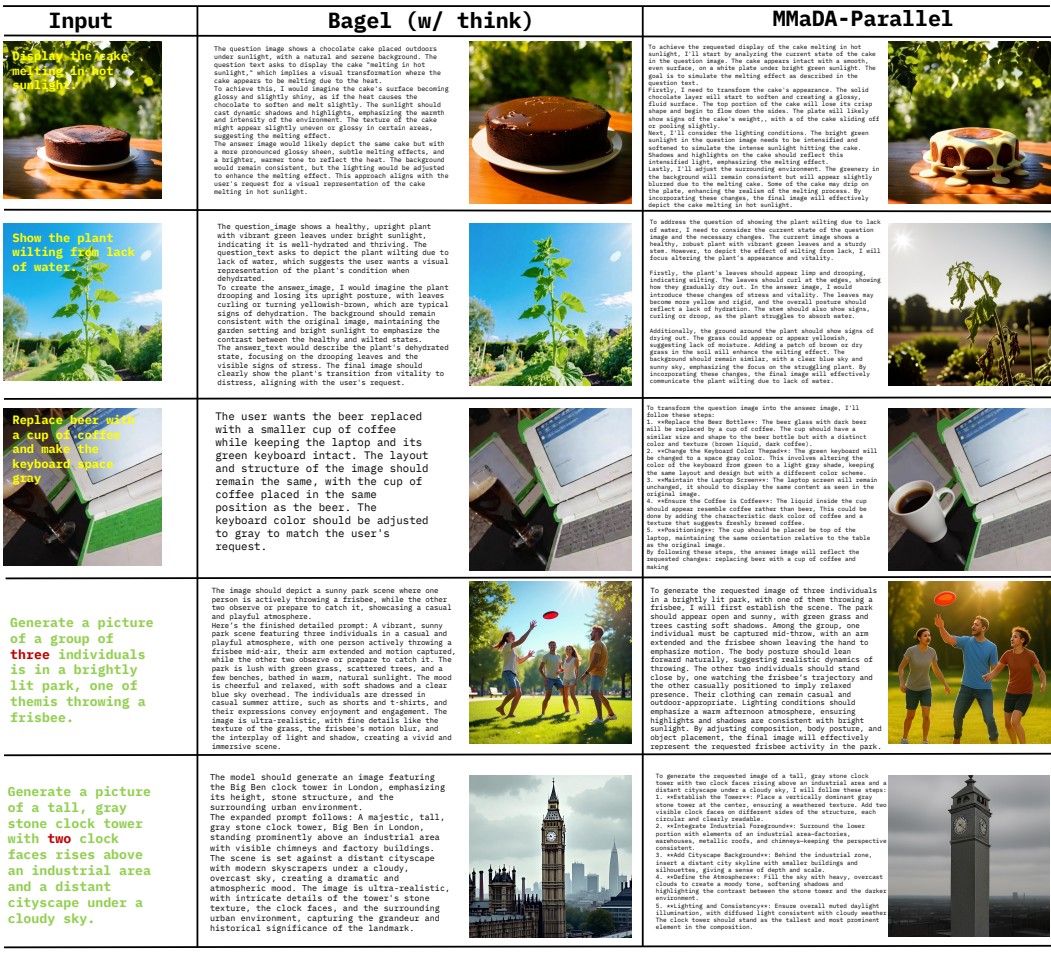

Figure 5: Qualitative results in comparison with Bagel.

models like GPT-4o and Gemini-2.5, **MMaDA-Parallel** substantially narrows the gap in alignment metrics while maintaining competitive text and image quality, demonstrating remarkable data efficiency. Furthermore, the results indicate that our ParaRL stage consistently improves output text-image consistency, suggesting that trajectory-level optimization effectively strengthens cross-modal grounding throughout the generation process.

In addition, we provide a qualitative comparison with open-source models in Figure 5, showcasing examples of both editing and generation. A key observation is that **MMaDA-Parallel** produces more precise and descriptive reasoning traces. This enhanced reasoning leads to superior visual fidelity in the final image. For instance, our model accurately renders complex instructions like a "melting cake" and correctly applies causal reasoning to depict "withered grass." Moreover, **MMaDA-Parallel** demonstrates stronger compositional abilities, particularly in counting, correctly generating "three people" or "two faces of a clock" where Bagel often fails. In contrast, Bagel's reasoning in these challenging cases tends to be vague or omits crucial details, leading to inaccurate image synthesis. These results further underscore **MMaDA-Parallel**'s capability for advanced thinking-aware editing and generation, driven by better-aligned semantic information.

### 4.3 ANALYSIS OF KEY CONTRIBUTIONS

Table 3: Parallel vs sequential decoding.

| Denoising | Text Align. | Image Align. | Output Align. |
|---|---|---|---|
| Sequential | 70.6 | 56.1 | 48.9 |
| **Parallel** | **70.4** | **58.2** | **51.5** |

Table 4: Output vs trajectory-level RL.

| Model | Text Align. | Image Align. | Output Align. |
|---|---|---|---|
| before RL | 70.4 | 58.2 | 51.5 |
| w/ Output-level RL | 70.7 | 62.3 | 53.6 |
| **w/ ParaRL (Ours)** | **71** | **63.2** | **59.8** |

Table 5: Ablation on sampling steps $s$ in ParaRL.

| ParaRL $s$ | Text Qual. | Text Align. | Image Cons. | Image Align. | Image Qual. | Output Align. | Overall |
|---|---|---|---|---|---|---|---|
| Before RL | 76.5 | 70.4 | 70.5 | 58.2 | 80.5 | 51.5 | 67.9 |
| ParaRL $s$=2 | 77.9 | 70.3 | 71.5 | 62.8 | 80.7 | 53.6 | 68.6 |
| ParaRL ($s$=3)  (default) | 80.4 | **71.0** | **73.4** | 63.2 | **81.2** | **59.8** | **71.5** |
| ParaRL ($s$=4) | **80.5** | 70.8 | 73.2 | **63.5** | 80.8 | 58.7 | 71.3 |

After presenting the overall results, we now return to the two central research questions that motivated our work: **RQ1:** Does parallel denoising improve generation quality compared with sequential denoising? **RQ2:** Does trajectory-level finetuning improve over output-level finetuning?

**The Benefit of Parallel Decoding (RQ1).**   We compare our model against a sequential baseline (*MMaA-Sequential*) that generates text before images. During training, noise was applied to only one modality at a time to align with this sequential inference process. Table 3 shows our parallel framework substantially outperforms this baseline on key alignment metrics, with comparable text and image quality. This result validates our core hypothesis: simultaneous, interactive decoding is crucial for reducing error propagation and producing coherent multimodal outputs.

**The Benefit of Trajectory-Level Optimization (RQ2).**   We compare two reinforcement learning strategies: (i) *output-level RL*, where rewards are computed on the final generated sample, and (ii) our proposed *ParaRL* with trajectory-level finetuning, where rewards are aggregated across denoising steps. As shown in Table 4, trajectory-level optimization yields gains in text–image consistency and output alignment, and Figure 6 further shows that it enables more stable training dynamics.

Another key hyperparameter in this strategy is the number of sampled steps, $s$. We analyze its impact in Table 5 and report the training curve in Figure 7. We find that using $s = 3$ or $s = 4$ yields substantial improvements over $s = 2$, as a denser reward signal provides more stable guidance. We adopt $s = 3$ in the final configuration for the best balance between performance and efficiency.

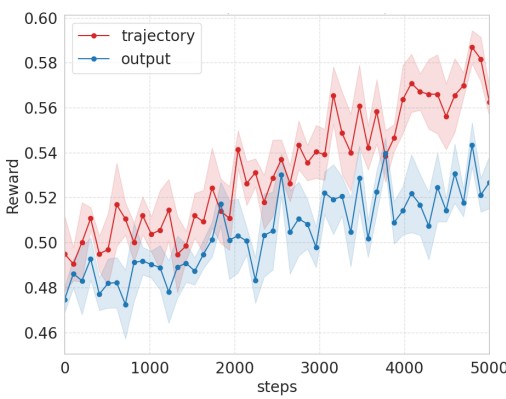

Figure 6: ParaRL reward training curve between trajectory and output level optimization.

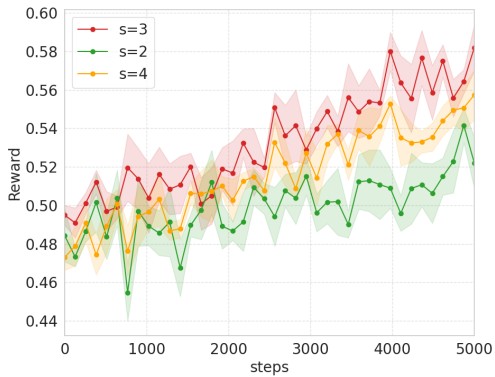

Figure 7: ParaRL reward training curve across different sampling steps of the trajectory.

## 5    CONCLUSION

In this work, we investigated a critical phenomenon where sequential thinking-aware models can paradoxically suffer from performance degradation on complex tasks. We conducted an in-depth analysis using our proposed ParaBench benchmark, which uniquely evaluates both output modalities, and found a strong correlation between this degradation and poor alignment between the generated modalities. To resolve this, we propose a parallel multimodal diffusion framework trained with supervised finetuning and further optimized by Parallel Reinforcement Learning (ParaRL)—our novel method of applying rewards along the entire denoising trajectory. Experiments validate that our approach significantly improves cross-modal alignment and semantic consistency, establishing a more robust paradigm for thinking-aware image synthesis.

**Acknowledgment.** This work is supported by the National Key Research and Development Program of China (No. 2023YFC3807600).

## ETHICS STATEMENT

This work advances research in text and image generation. We acknowledge that such models may be misused to create deceptive or harmful content, such as falsified images or misleading information. Our study is conducted for scientific purposes, and we encourage responsible use with appropriate safeguards to mitigate potential misuse.

## REPRODUCIBILITY STATEMENT

We provide detailed training implementation details in Appendix H and our main training code in the supplementary. All code and data are made public at MMaDA-Parallel-Code.

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

APPENDIX CONTENTS

## A  SCALING OF **MMADA-PARALLEL**

To further validate our **MMaDA-Parallel** on larger-scale training, we extend our post-training framework on Lumina-DiMOO Xin et al. (2025). Lumina-DiMOO shares a similar architecture with MMaDA, but benefits from much larger-scale data training and a substantially stronger visual tokenizer, amused-VQ Xin et al. (2025). The original MMaDA tokenizer is known to be a major bottleneck for visual fidelity and text rendering, which restricts the achievable performance of both sequential and parallel paradigms. By switching to the amused-VQ tokenizer, the limitations in reconstruction and fine-grained detail generation are largely removed, allowing us to evaluate our parallel framework in a setting where the tokenizer is no longer the dominant constraint. We adopt identical training settings as in Lumina-DiMOO, and report its corresponding quantitative and qualitative results in Table 6 and Figure 8. The results clearly show that after applying our Parallel framework and ParaRL post-training, Lumina-DiMOO surpasses BaGEL and achieves new state-of-the-art performance in thinking-aware synthesis. This strongly validates both the scalability and the headroom of our method once the tokenizer bottleneck is addressed.

| Input | Edit Prompt | Bagel (w/ think) | MMaDA-Parallel |
|---|---|---|---|
|  | Portray the flower **wilting** after being left in **hot sun**. |  |  |
|  | Illustrate the leaf **falling** from tree after **strong gust**. |  |  |
|  | Replace the laptops with **futuristic transparent** tablets displaying holographic screens, and change the drink to a cup of **glowing blue** energy drink. |  |  |
|  | Show the vase shattered on **floor** with **flowers scattered**. |  |  |
|  | Show the spiderweb **torn** by falling branch from nearby tree. |  |  |

Figure 8: Additional qualitative results on MMaDA-Parallel-A, post-trained from Lumina-DiMOO.

Table 6: **Main results on *ParaBench***. MMaDA-Parallel-A represents our variant post-trained from Lumina-DiMOO.

| Model | Text Qual. | Text Align. | Image Cons. | Image Align. | Image Qual. | Output Align. | Overall |
|---|---|---|---|---|---|---|---|
| **Open-source models (Non-thinking)** | | | | | | | |
| Flux.1-Dev | - | - | - | 65.2 | 77.5 | - | - |
| Qwen-Image | - | - | - | 67.2 | **84.2** | - | - |
| Flux.1-Kontext | - | - | 77.9 | 65 | 84 | - | - |
| Qwen-Image-Edit | - | - | **78.2** | **73.5** | 84.1 | - | - |
| Bagel (w/o think) | - | - | 72.2 | 50.3 | 80.1 | - | - |
| **Closed-source models** | | | | | | | |
| GPT-4o | 92.5 | 93.4 | 86.2 | **85.7** | 88.1 | **89.5** | **89.2** |
| Gemini-2.5 | 94.1 | 95.2 | 88.5 | 76.2 | **90.2** | 83.4 | 88.9 |
| **Open-source models (Thinking-aware)** | | | | | | | |
| Bagel (w/ think) | 82.0 | 74.5 | **76.7** | 63.4 | 81.5 | 52.9 | 71.8 |
| Show-o* (tuned) | 75.2 | 70.7 | 69.1 | 57.5 | 78.5 | 48.9 | 66.6 |
| **MMaDA-Parallel-A w/o Para-RL** | 82.6 | 73.7 | 71.3 | 64.6 | 82.6 | 63.3 | 73.0 |
| **MMaDA-Parallel-A w/ Para-RL** | **84.1** | **76.5** | 71.0 | **67.8** | **83.6** | **68.8** | **75.3** |

# B  ADDITIONAL RESULTS

## B.1  QUALITATIVE RESULTS

We provide more qualitative results in Figure 9 and Figure 10 for thinking-aware image editing and generation.

## B.2  QUANTITATIVE RESULTS

We also report additional image-only results of **MMaDA-Parallel** on RISEBench Wu et al. (2025c) in Table 7 and GenEval Ghosh et al. (2023) 8. Along with general image synthesis benchmarks

| Input Image | Prompt | Thinking-Aware Editing with MMaDA-Parallel |
|---|---|---|
|  | Draw what it will look like an hour later. | To address the question, we need to depict it an hour later, we'll focus on the current state shown in the question image. The goal is to simulate its appearance of several hours after passed. Observing the question image, the apple is partially opened with a vibrant interior and a central pit. To achieve the desired effect, we will need to adjust the lighting and shadows.\n\nAs the hour progresses, the lighting will soften, and the colors will blend more The outer skin of the apple will appear more translucent, possibly revealing more of its yellowish hue. The pit, which is currently less illuminated, should become more pronounced due to the light exposure. Additionally, the shadow cast by the apple will dark slightly, enhancing the sense of time and depth.\n\nTo achieve this, the we should gradually reduce the brightness and contrast of the apple's skin, making it a more muted and even tone. The pit should be made more exposed and detailed, showcasing its natural texture. The, the shadow should be adjusted to suggest the passage of time, with a slight fading toward the darkness. This will create a realistic depiction of the apple after an hour,, with a more texture and color changes, as seen in the answer image.  |
|  | What will happen after one hour? | To address the question, I need to consider what changes to the scene after an hour sets. Observing the initial image, we have a glass with iced tea and scattered seeds floating on the surface. To transition to the desired outcome, I'll need consider how the tea's appearance and as time approaches. \n\nAs the tea sets, the liquid will gradually thicken, taking on a more uniform hue. The ice cubes may start to float more prominently and become less distinct due to the warmth of the liquid. The seeds on the surface will also change, scattering more unevenly. Some seeds may appear larger and more prominent due to the increased water content. Additionally, the overall clarity of the tea will decrease, leading to a softer, more diffuse appearance seen in the second image.\n\nTo create this final image, I'll focus on adjusting the lighting and shadows to reflect the setting sun.. The glass should remain centered, but the background should subtly shift to suggest the passage of time. The seeds on the surface will be refined to show signs of settling and scattering, emphasizing the the progression of the tea process. This adjustment can effectively illustrate the creation of the scene an hour later, as depicted in the intended image.  |
|  | Draw an image showing a distant view of the trees from the same angle, farther away. | To create the image image showing a distant view of the trees from the same angle farther away, I'll start by analyzing the current composition. The question image features a single, prominent tree in the foreground with a clear blue sky and scattered clouds. The goal is to extend this view and reducing the distance while maintaining the same, ensuring the trees appear farther away.\n\nFirstly, I need to expand the tree in the scene. This means reducing its size relative to the viewer, making it appear smaller and more distant. The sky should remain consistent but become less dominant as the tree stretches out. \n\nNext, I'll adjust the perspective slightly. The current image shows a slight tilt, suggesting the camera is approaching the tree. To align with the idea of the trees being seen farther away, I will subtly blur the horizon and background. This blur will create a sense of depth, making the trees appear even more away.\n\nLastly, I'll consider adding subtle details such as faint outlines of additional trees or a horizon line in the distance, consistent with the broader perspective.  |
|  | What will happen if he releases his hand? | To address the question, "What will happen if he releases his hand?" I need to consider the dynamics depicted in the question image. In the current image, the see a hand is positioned over a stack of objects, with the hand resting on top of a stack of stones. The question implies a scenario where the hand is released, suggesting a change in the arrangement of the objects.\n\nTo visualize this, I\'ll focus on the dynamics of the stacked objects. As the hand is removed from the stack, it likely will shift upward. The hand should no longer be resting on top of the stack, and the top object will begin to float away. The orange objects at the bottom of the stack will remain in their position but may shift slightly due to the change in the stack.\n\nTo create the answer image, I would imagine the hand being removed from the stack, with the top object lifted and floating away. The orange object at the bottom of the stack will stay in place, but its position will slightly adjust to reflect the release of the hand. Additionally, the background could subtly lighting to emphasize the connection between the hand and the floating objects, maintaining the continuity of the scene. This approach aligns with the question of what happens when the hand is released  |
|  | Adjust the small hamburger to the size of the large hamburger | To adjust the small hamburger to the size of the large hamburger, I'll start by analyzing the current differences between the two images. In the question image, the small hamburger appears significantly smaller compared to the large one. The goal is to make the small hamburger match the scale and proportions of the large one.\n\nFirst, I'll focus on reducing the size of the small hamburger. This involves scalingizing its height and width so that it appears smaller in relation to the large hamburger. To achieve this, I'll increase the size of the small burger proportionally it is, maintaining the same level of detail as the large burger.\n\nNext, I'll consider the toppings adjustment. The small burger already has a layer of lettuce, tomato, and cheese, which need to be adjusted slightly to align with the large burger. This involves adjusting the proportions of these ingredients to create a balanced appearance on the small burger.\n\nFinally, I'll ensure the background remains consistent between both images. The wooden surface and the warm lighting should should remain unchanged, creating a cohesive visual flow between the two images.\n\nBy making these adjustments, the final image will show the small hamburger harmoniously integrated with the large hamburger, matching the proportions as shown in the original image.  |

Figure 9: Additional qualitative results on thinking-aware image editing.

WISE Niu et al. (2025) and Gedit-Bench Liu et al. (2025c). For fair comparison, we evaluate against the sequential version of MMaDA, MMaDA-Parallel. This variant of MMaDA was trained using the same data and resources as our **MMaDA-Parallel**, but trained and inferred in a sequential manner. We find that **MMaDA-Parallel** achieves consistent performance improvements, demonstrating that parallel generation leads to overall gains in image alignment. Compared with the original MMaDA, our approach further narrows the performance gap with Bagel.

## C    MORE RELATED WORK

**Diffusion large language models.**    Diffusion models have achieved remarkable progress in vision (Ho et al., 2020; Rombach et al., 2022; Esser et al., 2024b; Song et al., 2020; Peebles & Xie, 2023), motivating their extension to text. The discrete nature of textual tokens, however, makes direct adaptation non-trivial. Two main approaches have emerged: learning continuous latent representations (Chen et al., 2022; Mahabadi et al., 2023; Ye et al., 2023; Gong et al., 2022), and designing discrete diffusion models (Ou et al., 2024; Gong et al., 2024; Liu et al., 2025d; Ye et al.,

| Input Prompt | Thinking-Aware Generation with MMaDA-Parallel |
|---|---|
| A Starbucks coffee cup and bread rolls on a table next to a window with a black car parked outside. | To generate the image based on the prompt "A Starbucks coffee cup and bread rolls on a table next to a window with a black car parked outside", I need to focus on the following elements: 1. Place a Starbucks coffee cup prominently in the foreground on the left side of the frame. 2. Position a plate with bread rolls on the right side of the frame. 3. Ensure the table is wooden and has a warm tone. 4. Include a window in the background with a view of a street. 5. Have a black car parked outside visible through the window. 6. Set the scene during daylight hours to capture natural light. 7. Ensure the overall composition is balanced with the coffee cup and bread rolls as focal points. This will create a realistic and appealing image that aligns with the given prompt. |
| A man sitting comfortably on a wooden bench in the midst of a tranquil forest glade, encircled by towering trees. | To generate the image for the prompt "A man sitting comfortably on a wooden bench in the midst of a tranquil forest glade, encircled by towering trees," I need to focus on the following key elements: 1. **Main Subject**: A man sitting comfortably on a wooden bench. 2. **Setting**: A tranquil forest glade. 3. **Atmosphere**: Tranquil and serene. 4. **Details**: Wooden bench, tall trees, and a peaceful environment. The composition should place the man as the focal point, surrounded by the forest, with the bench being the central element. The background should feature tall trees and greenery to enhance the sense of being in a forest glade. The lighting should be soft and natural, suggesting a calm, early morning or late afternoon setting. The overall color palette should be earthy and green, with muted tones to match the tranquil atmosphere. |
| A trio of chairs, a trio of banners, a trio of desks, a quartet of ships, and a quartet of cushions. | To generate the image based on the prompt "A trio of chairs, a trio of banners, a trio of desks, a quartet of ships, and a quartet of cushions," I need to focus on creating a balanced composition with multiple groups of objects. Here's the planning process: 1. **Chairs**: Place three distinct types of chairs (e.g., a wooden chair, a metal chair, and a fabric chair) in the foreground, each with unique textures and colors. 2. **Banners**: Include three different banners, perhaps with various messages or designs, hung in a semi-circle above the chairs to add depth and interest. 3. **Desks**: Position three desks of varying sizes and styles behind the chairs, ensuring they are spaced out to maintain balance. 4. **Ships**: Introduce four ships, possibly toy models or detailed illustrations, scattered around the scene, adding a nautical element. 5. **Cushions**: Add four cushions in soft, contrasting colors, placed near the desks to enhance the cozy atmosphere. The overall composition should be vibrant and varied, with each group of objects clearly distinguishable yet harmoniously integrated into the scene. The lighting should be even to highlight the details of each object.</think> To generate the image, I would start by sketching the layout, ensuring there is enough space for all the objects. Then, I would choose appropriate colors and textures for each item. After that, I would render the scene with careful attention to lighting and perspective to bring the image to life. |
| Create an image featuring three differently colored baseball gloves arranged on a wooden bench. | To generate an image featuring three differently colored baseball gloves arranged on a wooden bench, I would start by choosing vibrant colors for the gloves, such as orange, black, and a light brown. Next, I would select a realistic wooden bench with a natural grain pattern to provide texture and contrast against the smooth leather of the gloves. The arrangement should showcase the gloves side by side, with their open palms facing forward, allowing viewers to see the detailed stitching and laces. The lighting should be soft and even to enhance the colors and textures without creating harsh shadows. Finally, the background could be minimal, perhaps a blurred outdoor setting like a park bench, to keep the focus on the gloves. |
| A plate of pancakes topped with butter, bacon strips, a side of syrup, and a coffee cup on a table in a diner setting. | To generate the image based on the prompt "A plate of pancakes topped with butter, bacon strips, a side of syrup, and a coffee cup on a table in a diner setting," I need to plan the following visual elements: 1. **Main Subject**: A plate of pancakes with toppings (butter, bacon strips, and syrup). 2. **Side Item**: A coffee cup placed next to the plate. 3. **Setting**: A diner environment, which includes a table and typical diner decor. 4. **Lighting**: Soft, natural light to mimic a cozy diner atmosphere. 5. **Composition**: Center the plate of pancakes in the foreground, with the coffee cup slightly off-center for balance. 6. **Background**: Show a diner interior with tables, chairs, and possibly some patrons in the distance to establish the setting. The overall goal is to create a warm, inviting, and appetizing image that captures the essence of a classic diner breakfast scene. |

Figure 10: Additional qualitative results on thinking-aware image generation.

Table 7: **Overall performance on RISEBench.** .

| Models | Temporal | Causal | Spatial | Logical | Overall |
|---|---|---|---|---|---|
| GPT-4o-Image | **34.1%** | **32.2%** | **37.0%** | **10.6%** | **28.9%** |
| Gemini-2.0-Flash-exp | 8.2% | 15.5% | 23.0% | 4.7% | 13.3% |
| Bagel | 3.5% | 4.4% | 9.0% | 5.9% | 5.8% |
| MMaDA(Sequential) | 3.9 % | 5.2% | 8.1% | 4.8% | 5.5% |
| MMaDA-Parallel | 4.2% | 5.5% | 8.3% | 5.1% | 5.75% |

2025b; Zhu et al., 2025). Among the latter, **Masked Diffusion Models (MDMs)** stand out by leveraging bidirectional attention for global consistency and supporting parallel decoding. Systems such as Dream7B (Ye et al., 2025b) and LLaDA (Nie et al., 2025) achieve performance comparable to autoregressive LLMs. Beyond text, diffusion-based LLMs have also been extended to multimodal

Table 8: **Results on GenEval.**.

| Method | Single Obj. | Two Obj. | Counting | Colors | Position | Color Attri. | Overall |
|---|---|---|---|---|---|---|---|
| SDXL | **0.98** | 0.74 | 0.39 | **0.85** | 0.15 | 0.23 | 0.55 |
| Show-o Xie et al. (2024) | 0.95 | 0.52 | 0.49 | 0.82 | 0.11 | 0.28 | 0.53 |
| MMaDA (Yang et al., 2025a) | 0.99 | 0.76 | 0.61 | 0.84 | 0.20 | 0.37 | 0.63 |
| Bagel (Deng et al., 2025a) | 0.98 | 0.95 | 0.84 | 0.95 | 0.78 | 0.77 | 0.88 |
| MMaDA(Sequential) | 0.99 | 0.78 | 0.66 | 0.87 | 0.34 | 0.37 | 0.68 |
| MMaDA-Parallel | 0.99 | 0.83 | 0.70 | 0.88 | 0.40 | 0.47 | 0.71 |

Table 9: Results on WISE

| Model | Cultural | Time | Space | Biology | Physics | Chemistry | Overall |
|---|---|---|---|---|---|---|---|
| SDXL | 0.43 | 0.48 | 0.47 | 0.44 | 0.45 | 0.27 | 0.43 |
| Show-o Xie et al. (2024) | 0.28 | 0.36 | 0.40 | 0.23 | 0.33 | 0.22 | 0.30 |
| Bagel Deng et al. (2025a) | 0.44 | 0.55 | 0.68 | 0.44 | 0.60 | 0.39 | 0.52 |
| MMaDA-Sequential | 0.39 | 0.54 | 0.58 | 0.55 | 0.44 | 0.22 | 0.44 |
| **MMaDA-Parallel** | 0.42 | 0.56 | 0.59 | 0.57 | 0.47 | 0.24 | 0.47 |

domains. LaViDA (Li et al., 2025) employs multi-view image encoding with masked-denoising training, LLaDA-V (You et al., 2025) integrates masked diffusion with visual instruction tuning, and MMaDA (Yang et al., 2025a) unifies reasoning across text and vision generation through chain-of-thought supervision and reinforcement learning. These advances highlight the scalability and versatility of diffusion-based language models across both unimodal and multimodal settings. Nevertheless, existing approaches have not yet explored **parallel text–image co-generation**, leaving cross-modal reasoning and alignment still constrained by sequential pipelines.

**Reinforcement learning for multimodal foundation models.** Reinforcement Learning (RL) has emerged as a powerful paradigm for enhancing reasoning and controllability in large models. The widely adopted GRPO (Guo et al., 2025a) applies rewards primarily on the correctness of the final answer and the adherence to a predefined format. Recently, RL has been adopted in multimodal large language models (Chen et al., 2025b; Meng et al., 2025; Yang et al., 2025b; Zhang et al., 2025; Deng et al., 2025b; Huang et al., 2025b), incorporating task-specific rewards such as answer correctness, intersection-over-union (IoU) for localization (Liu et al., 2025e), and image–text alignment scores (e.g., T2I-R1 (Jiang et al., 2025a)). Extensions such as (Jiang et al., 2025b; Hong et al., 2025) further introduce cross-modality coherence rewards. In the context of diffusion language models, similar strategies have been explored with verified rewards and carefully designed probability approximations (Yang et al., 2025a; Gong et al., 2025) . Despite these advances, most existing methods focus solely on rewards applied to the final output, while largely ignoring the generative trajectory. This overlooks the fact that intermediate steps can provide crucial signals for alignment. In contrast, our work investigates the synergy between modalities during the denoising process and introduces ParaRL, which exploits stepwise semantic alignment to optimize thinking-aware multimodal generation.

# D PRELIMINARIES

## D.1 PRELIMINARIES OF DISCRETE DIFFUSION MODELS.

In recent years, diffusion models have set new standards in generative modeling. While Denoising Diffusion Probabilistic Models (DDPMs) excel in continuous domains like raw pixel spaces, Discrete Denoising Diffusion Probabilistic Models (D3PMs) have proven highly effective for discrete data, such as tokenized images and text. Models like VQ-Diffusion Gu et al. (2022), MaskGIT (Chang et al., 2022), Muse (Chang et al., 2023), Show-o (Xie et al., 2024), and MMaDA Yang et al. (2025a) have demonstrated that a discrete diffusion process can generate high-fidelity outputs with great efficiency. Our model's architecture is built upon this discrete diffusion paradigm. We now provide the formal preliminaries, beginning with the foundational forward and

Table 10: Results on GEdit-Bench

|  | G_SC | G_PQ | G_O |
|---|---|---|---|
| Gemini 2.0 | 6.73 | 6.61 | 6.32 |
| GPT-4o | **7.85** | **7.62** | **7.53** |
| Instruct-Pix2Pix (Brooks et al., 2023) | 3.58 | 5.49 | 3.68 |
| MagicBrush (Zhang et al., 2023) | 4.68 | 5.66 | 4.52 |
| AnyEdit (Yu et al., 2025) | 3.18 | 5.82 | 3.21 |
| Step1X-Edit Liu et al. (2025c) | 7.09 | 6.76 | 6.70 |
| Bagel Deng et al. (2025a) | 7.36 | 6.83 | 6.52 |
| MMaDA-Sequential | 5.63 | 5.97 | 5.13 |
| **MMaDA-Parallel** | **5.72** | **6.28** | **5.23** |

reverse processes and culminating in the simplified mask-and-predict training objective that our model employs.

**Forward and reverse processes.** A discrete diffusion model consists of two key processes: (1) The *Forward Process* ($q$), a fixed Markov chain that gradually corrupts input data $\mathbf{x}_0$ over $T$ timesteps into noisy latents $\mathbf{x}_1, \ldots, \mathbf{x}_T$; and (2) The *Reverse Process* ($p_\theta$), a learned neural network that reverses this corruption by progressively denoising $\mathbf{x}_T$ to recover the original data distribution. Let's consider a single token $x_0 \in \{1, \ldots, K\}$ from a codebook of size $K$. The forward process at each step $t$ is defined by a stochastic transition matrix $\mathbf{Q}_t \in \mathbb{R}^{K \times K}$. A key property is that the distribution of $\mathbf{x}_t$ conditioned on the initial state $\mathbf{x}_0$ is tractable:

$$q(\mathbf{x}_t|\mathbf{x}_0) = \text{Cat}(\mathbf{x}_t|\mathbf{x}_0\overline{\mathbf{Q}}_t), \quad \text{where} \quad \overline{\mathbf{Q}}_t = \mathbf{Q}_1\mathbf{Q}_2 \cdots \mathbf{Q}_t. \tag{5}$$

The posterior probability, which is essential for training, is also tractable:

$$q(\mathbf{x}_{t-1}|\mathbf{x}_t, \mathbf{x}_0) = \frac{q(\mathbf{x}_t|\mathbf{x}_{t-1})q(\mathbf{x}_{t-1}|\mathbf{x}_0)}{q(\mathbf{x}_t|\mathbf{x}_0)} \propto \text{Cat}\left(\mathbf{x}_{t-1}\left|\frac{\mathbf{x}_t\mathbf{Q}_t^\top \odot \mathbf{x}_0\overline{\mathbf{Q}}_{t-1}}{\mathbf{x}_0\overline{\mathbf{Q}}_t\mathbf{x}_t^\top}\right.\right), \tag{6}$$

where $\odot$ denotes element-wise product.

**Absorbing mask state and transition matrix.** The design of the transition matrix $\mathbf{Q}_t$ dictates the nature of the corruption. A highly effective approach, inspired by masked language modeling, is to introduce a special **absorbing [MASK] state**. This expands the token vocabulary to $K + 1$ states. Once a token becomes [MASK], it remains masked for all subsequent timesteps. This explicitly signals corrupted positions to the model. The transition matrix for this "Absorbing-Uniform" process is defined as:

$$\mathbf{Q}_t = \begin{bmatrix} \omega_t + \nu_t & \nu_t & \cdots & \nu_t & \alpha_t \\ \nu_t & \omega_t + \nu_t & \cdots & \nu_t & \alpha_t \\ \vdots & \vdots & \ddots & \vdots & \vdots \\ \nu_t & \nu_t & \cdots & \omega_t + \nu_t & \alpha_t \\ 0 & 0 & \cdots & 0 & 1 \end{bmatrix} \in \mathbb{R}^{(K+1)\times(K+1)}, \tag{7}$$

where at each step $t$, a token has a probability $\alpha_t$ to be masked, a probability $\beta_t$ to be replaced by a random token, and a probability $\omega_t = (1 - \alpha_t - \beta_t)$ to remain unchanged. The [MASK] token (last row) always transitions to itself.

**Objective as mask prediction.** The training objective for diffusion models is derived by maximizing the Evidence Lower Bound (ELBO) on the data log-likelihood. The negative ELBO, which is minimized during training, can be decomposed into several terms representing different stages of

the diffusion process:

$$-\mathcal{L}_{\text{ELBO}} = \mathbb{E}_q \Bigg[ \underbrace{- \log p_\theta(\mathbf{x}_0|\mathbf{x}_1)}_{\text{Reconstruction Term}} + \sum_{t=2}^{T} \underbrace{\text{KL}(q(\mathbf{x}_{t-1}|\mathbf{x}_t, \mathbf{x}_0) \| p_\theta(\mathbf{x}_{t-1}|\mathbf{x}_t))}_{\text{Denoising Matching}}$$
$$+ \underbrace{\text{KL}(q(\mathbf{x}_T|\mathbf{x}_0) \| p(\mathbf{x}_T))}_{\text{Prior Matching}} \Bigg]. \tag{8}$$

Here, the objective consists of three main components: (1) a reconstruction term that learns to generate the final data from $\mathbf{x}_1$, (2) a series of KL divergence terms that train the reverse process $p_\theta$ to match the true posterior at each denoising step, and (3) a prior matching term that aligns the final noisy latent with a simple prior distribution. Following derivations in D3PMs Austin et al. (2021), this complex objective can be simplified to a weighted sum of reconstruction terms:

$$\mathcal{L}_{\text{simple}} = \sum_{t=1}^{T} \mathbb{E}_{q(\mathbf{x}_0, \mathbf{x}_t)}[-\log p_\theta(\mathbf{x}_0|\mathbf{x}_t)]. \tag{9}$$

When using the absorbing mask state strategy, this simplified objective becomes equivalent to a **Cross-Entropy loss** for mask token prediction, as used in MaskGIT Chang et al. (2022). This approach is highly effective as it focuses the model's capacity on reconstructing only the corrupted parts of the data. Our work leverages this powerful paradigm for both text and image token generation.

## D.2 GROUP RELATIVE POLICY OPTIMIZATION FOR DISCRETE DIFFUSION MODELS

Group Relative Policy Optimization (GRPO) (Guo et al., 2025a) is a powerful policy gradient algorithm originally designed for autoregressive models. However, its direct application to discrete diffusion models is non-trivial. The core challenge lies in computing the importance sampling ratios and sequence-level likelihoods; these are straightforward in an autoregressive chain but ill-defined in a non-autoregressive, parallel decoding process. Diffusion models lack a sequential history for token-level probabilities, and their policy distributions are implicitly dependent on masking patterns, making direct likelihood estimation computationally prohibitive.

To bridge this gap, we adopt the efficient random masking framework from MMaDA (Yang et al., 2025a) to adapt GRPO for our diffusion-based architecture. This strategy circumvents the need for direct likelihood computation by using the model's predictions on randomly masked inputs as an unbiased estimate of the policy likelihoods. First, the advantage $\hat{A}_i$ for each response $o_i$ in a generated group $\{o_j\}_{j=1}^{G}$ is computed in the standard group-relative manner:

$$\hat{A}_i = \frac{r_i - \text{mean}(\{r_j\}_{j=1}^{G})}{\text{std}(\{r_j\}_{j=1}^{G}) + \epsilon}, \tag{10}$$

where $r_i$ is the reward for response $o_i$. The policy gradient is then calculated using an importance sampling ratio $r'_{i,t}(\theta)$ defined over a randomly masked version of each response, where a unique mask ratio $p_i \sim U[0, 1]$ is sampled for each response at each training step. This allows the standard clipped GRPO objective to be adapted for diffusion models as follows:

$$\mathcal{J}_{\text{Diff-GRPO}}(\theta) = \mathbb{E}_{\substack{q \sim \mathcal{D}, \{o_i\} \sim \pi_{\text{old}}, \\ \{p_i\} \sim U[0,1]}} \Bigg[ \frac{1}{G} \sum_{i=1}^{G} \frac{1}{|\mathbf{M}_i|} \sum_{t \in \mathbf{M}_i} \bigg( \min \Big( r'_{i,t}(\theta) \hat{A}_i, $$
$$\text{clip}\Big( r'_{i,t}(\theta), 1 - \varepsilon, 1 + \varepsilon \Big) \hat{A}_i \Big) \bigg) - \beta D_{\text{KL}}(\pi'_\theta \| \pi'_{\text{ref}}) \Bigg], \tag{11}$$

where the expectation is also taken over the random mask ratios, the inner summation is only over the masked tokens $\mathbf{M}_i$, and $\pi'$ denotes the policy likelihoods approximated via the masking scheme. This formulation enables stable and efficient policy optimization by effectively adapting the principles of GRPO to a non-autoregressive setting.

# E   SAMPLING DETAILS ON TEXT AND IMAGE

**Parallel sampling and denoising strategy.**   Our model employs a parallel sampling strategy, predicting logits for all text and image tokens simultaneously in a single forward pass. The denoising process for both modalities is guided by a confidence-based re-masking schedule, inspired by MaskGIT (Chang et al., 2022) and LLaDA (Nie et al., 2025). Crucially, while the logits are generated jointly, we apply distinct masking schedulers and confidence metrics to the text and image tokens to account for their different statistical properties and generation requirements.

**Image token denoising.**   For image generation, we follow the iterative decoding process from MaskGIT. At each timestep $t$, given the current set of $M$ masked image tokens, the model predicts logits $\ell^t = \{\ell_i^t\}_{i=1}^M$. For each masked position $i$, we sample a candidate token $u_i'$ from the predicted probability distribution and compute its confidence score $s_i$. A mask scheduling function $\gamma(t/T)$ determines the number of tokens $m = \lceil \gamma(t/T)M \rceil$ that should be kept (i.e., remain unmasked). We select the $m$ tokens with the highest confidence scores to keep for the next step $t+1$, and the remaining $M - m$ tokens are re-masked. The update rule for a token at position $i$ is:

$$u_i^{(t+1)} = \begin{cases} u_*, & \text{if } s_i < \text{sorted}_j(s_j)[m] \\ u_i', & \text{otherwise} \end{cases},$$
(12)

where $u_*$ represents the [MASK] token and $\text{sorted}_j(s_j)[m]$ is the $m$-th value in the sorted list of confidence scores. This iterative refinement continues until all image tokens are finalized. In our implementation, we generate a 512px image, which is encoded into 1024 discrete tokens and takes 30 steps to decode.

**Text token denoising.**   For text generation, we adopt the semi-autoregressive denoising strategy from LLaDA (Nie et al., 2025), where the output sequence is generated in blocks from left to right. Within each block, however, generation is non-autoregressive and iterative. The core of this process is a reverse sampling step that transforms a partially masked sequence $\mathbf{x}_t$ at step $t$ into a less masked sequence $\mathbf{x}_s$ at an earlier step $s < t$. This transition is formally characterized by the probability:

$$q_{s|t}(\mathbf{x}_s|\mathbf{x}_t) = \prod_{i=0}^{N-1} q_{s|t}(x_s^i|\mathbf{x}_t^i) \quad \text{and} \quad q_{s|t}(x_s^i|\mathbf{x}_t^i) = \begin{cases} 1, & x_t^i \neq \text{[M]}, x_s^i = x_t^i \\ \frac{1}{1-\alpha_t}, & x_t^i = \text{[M]}, x_s^i = \text{[M]} \\ \frac{\alpha_s - \alpha_t}{1-\alpha_t} p_\theta(x_0^i|\mathbf{x}_t), & x_t^i = \text{[M]}, x_s^i \neq \text{[M]} \\ 0, & \text{otherwise}, \end{cases}$$
(13)

where $p_\theta(x_0^i|\mathbf{x}_t)$ is the model's prediction of the original token for the masked position $i$ and $\alpha_t = 1 - t$. In practice, this involves an iterative refinement loop. At each step, given the current sequence $\mathbf{x}_t$, we first sample candidate tokens for all masked positions. Then, following the deterministic low-confidence re-masking strategy adopted by LLaDA, we identify the tokens with the lowest prediction confidence scores and re-mask them for the next refinement iteration.

In our implementation, we generate the sequence with 256 sequence length, in blocks of 64 tokens and 128 steps. At each denoising step within a block, we unmask the two tokens with the lowest confidence scores. This block-based, semi-autoregressive approach is essential for generating coherent and naturally structured sentences, as it mitigates issues like the premature generation of end-of-sequence (|EOS|) tokens that can arise in a fully non-autoregressive setting.

# F   DETAILS OF TRAINING DATASET CURATION

Our training dataset is a carefully curated collection of 150,000 high-quality samples designed for thinking-aware image synthesis. The primary challenge was that existing public datasets for image editing and generation typically provide input-output pairs without the intermediate reasoning traces required by our method. Therefore, our curation process involved three main stages: (1) aggregating data from state-of-the-art sources, (2) generating high-quality reasoning traces to augment this data, and (3) applying a rigorous filtering and enhancement pipeline. The final dataset consists of 100,000 editing pairs and 50,000 generation pairs, achieving a 2:1 ratio. An overview of the dataset is shown in Figure 11 and  12

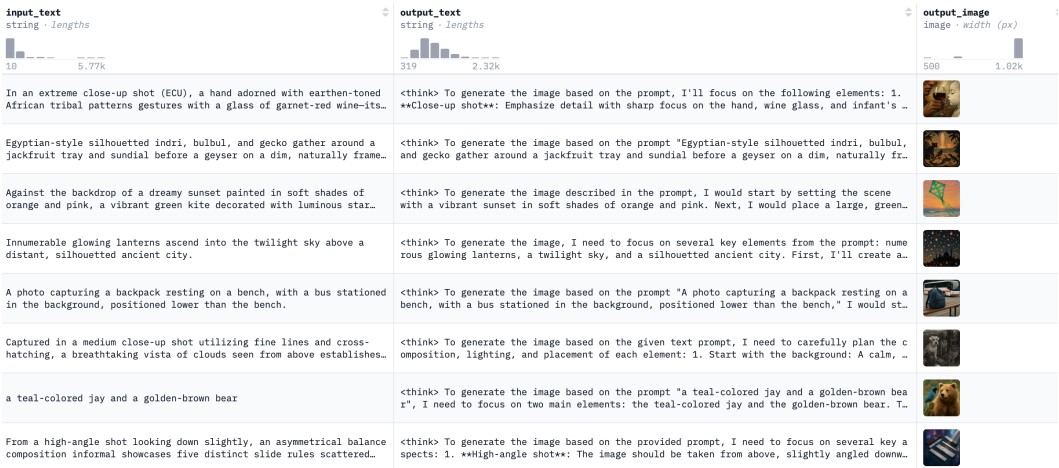

Figure 11: Overview of our dataset for thinking-aware editing

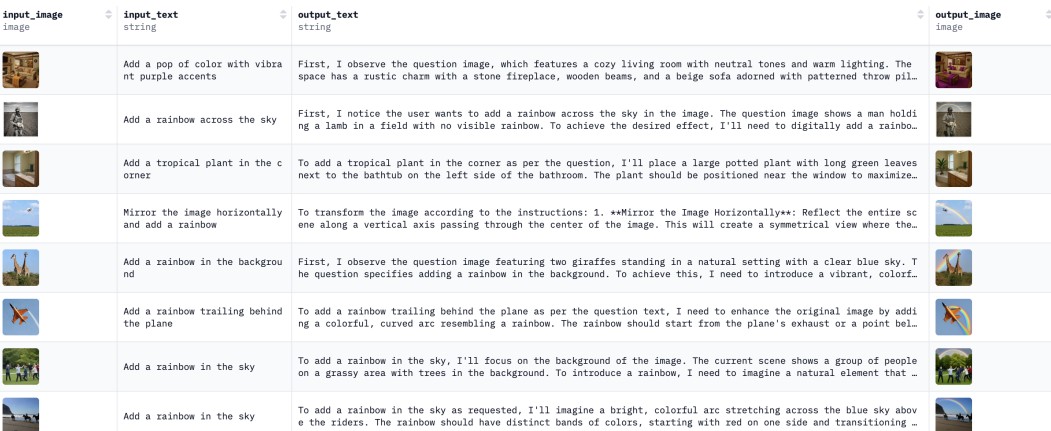

Figure 12: Overview of our dataset for thinking-aware editing

**Source datasets for editing data.** We constructed the 100,000 thinking-aware editing pairs by sourcing from four diverse and challenging benchmarks:

- **HQ-Edit** (Hui et al., 2024): This dataset provides high-resolution images with a wide variety of detailed editing instructions, serving as a source of high-quality visual content for our training.

- **UltraEdit** (Zhao et al., 2024): We leverage UltraEdit for its collection of complex editing instructions that require strong reasoning and compositional abilities, pushing the model beyond simple object manipulation.

- **AnyEdit** (Yu et al., 2025): Given the vast size of AnyEdit, we selectively sampled from its more challenging categories. Specifically, we focused on the `implicit_editing` subset, which contains instructions that do not explicitly mention the target object, requiring the model to infer the user's intent.

- **EditWorld** (Yang et al., 2024): This dataset is crucial for its focus on edits that require world knowledge and complex reasoning, such as causal (e.g., "what if a storm occurs") and temporal (e.g., "What's this man like in twenty years?") edits. To further bolster our model's capabilities in these areas, we performed data augmentation on this subset, using GPT-4o to generate three times the amount of similar, complex reasoning-based instructions and corresponding edits.

**Source dataset for generation Data.** For the 50,000 thinking-aware generation pairs, we sourced data from **ShareGPT4o** (Chen et al., 2025a). This dataset contains a rich collection of diverse, real-world prompts and corresponding high-quality image outputs, providing a strong foundation for general-purpose, knowledge-intensive image synthesis.

**Reasoning trace generation.** A core step in our curation process was to augment the source data with reasoning traces. Since the original datasets only provide triplets of ('input image', 'instruction', 'output image'), we utilized the powerful multimodal model **Qwen2.5-VL-7B** (Bai et al., 2025) to generate a plausible reasoning text for each sample. The model was prompted with the input/output image pair and the instruction to produce a step-by-step rationale explaining the transformation. This transformed our dataset into quadruplets: ('input image', 'instruction', 'reasoning trace', 'output image'), which is the required format for our thinking-aware training.

**Data filtering and quality control.** Finally, to ensure the highest quality, we applied a multi-stage filtering pipeline to the entire 150,000-sample dataset. First, we removed near-duplicates to increase data diversity. Second, we used a scoring mechanism based on Qwen-VL to identify and discard samples with low-quality or visually unappealing images. For cases where the instruction was valuable but the image quality was poor, we leveraged **GPT-4o** to regenerate higher-fidelity candidate images. This comprehensive curation process resulted in a clean, diverse, and high-quality dataset optimized for our training objectives.

## G  DETAILS OF PARABENCH

ParaBench is a comprehensive benchmark designed to address the limitations of existing evaluation protocols for thinking-aware image synthesis. Unlike traditional benchmarks that focus solely on the final image, ParaBench is built to assess the entire generation process, including the quality of the intermediate reasoning trace and its synergy with the visual output. It comprises a total of 300 challenging prompts, curated from various sources and divided into 200 for editing and 100 for generation.

**Composition of editing prompts.** The 200 editing prompts are meticulously curated and synthesized from various existing benchmarks to test a wide spectrum of complex reasoning abilities. To provide a structured analysis, we group them into five distinct categories:

- **Spatial Reasoning (40 prompts):** These are tasks requiring a deep understanding of object locations, orientations, and spatial relationships. Examples include instructions like "place the book to the left of the lamp" or "make the person in the background larger."
- **Temporal Reasoning (40 prompts):** These prompts involve reasoning about time and require the model to infer past or future states. Examples include "show what this street might look like 50 years from now" or "revert the shattered vase to its original state."
- **Causal Reasoning (40 prompts):** This category contains instructions that require the model to infer and depict cause-and-effect relationships. Examples include "show the ground after a heavy rain" or "make the plants look like they haven't been watered for weeks."
- **World Knowledge (40 prompts):** These are edits that require external, real-world knowledge to execute correctly. Examples include instructions like "turn this car into a model from the 1980s" or "edit the painting to be in the style of Van Gogh."
- **General Editing (40 prompts):** This category includes a broad set of common, foundational editing operations that do not fit into the specialized categories above. It primarily consists of instructions for adding, removing, or replacing objects and serves as a baseline for fundamental editing capabilities.

**Composition of generation prompts.** The 100 generation prompts are sourced from the ShareGPT4o (Chen et al., 2025a) dataset. They are designed to be open-ended and cover a wide range of scenarios, including the generation of creative scenes, complex compositions with multiple interacting objects, and images that require interpreting long, descriptive narratives.

**Evaluation axes.** All 300 prompts in ParaBench are evaluated using our LLM-as-a-judge framework across six fine-grained axes to provide a holistic assessment of a model's performance. The evaluation criteria are as follows:

- **Text Quality:** Assesses the fluency, coherence, and grammatical correctness of the generated reasoning text.
- **Text Alignment:** Measures how well the reasoning text follows the user's input instruction and accurately plans the edit/generation.
- **Image Quality:** Evaluates the photorealism, aesthetic quality, and absence of visual artifacts in the generated image.
- **Image Alignment:** Measures how faithfully the generated image adheres to the user's instruction.
- **Image Consistency (for editing tasks):** Assesses how well the model preserves the unedited parts of the original image, maintaining background, style, and object identity.
- **Output Alignment:** Evaluates the cross-modal consistency between the generated reasoning text and the final generated image.

We provide the prompts for thinking-aware image editing in Appendix M.The prompts for image generation follow the same format, with only minor modifications in the input and representation style.

## H  MORE IMPLEMENTATION DETAILS

**Training details.** Our model is initialized from the weights of MMaDA-MixCoT (Yang et al., 2025a), which utilizes LLaDA-8B as its text backbone and MagVIT-v2 for image tokenization. The post-training process consists of two stages. In the first stage, we perform supervised finetuning (SFT) for 30,000 steps on our curated dataset of 150,000 thinking-aware samples. In the second stage, we conduct Parallel Reinforcement Learning (ParaRL) for 10,000 steps, using a challenging subset of approximately 15,000 examples (10%) drawn from the SFT dataset. Both training stages were conducted on 32 NVIDIA A100 GPUs with a global batch size of 768. We utilized the AdamW optimizer with a learning rate of 2e-5 and a cosine learning rate schedule with a warm-up of 500 steps. We drop 10% of text input and 10% of image input to support classifier-free guidance sampling.

In ParaRL, we randomly sample $s = 3$ trajectory points. The steps of these certain points are identical in the same rollout and uniformly sampled in all rollouts. We set KL constraints $\beta = 0.0001$ to keep the same with MMaDA's baseline.

**Inference details.** During inference, our model employs a parallel sampling strategy, generating the logits for all text and image tokens simultaneously in a single forward pass. The images are generated with classifier-free guidance scale of 3.5, and text with a scale of 0.

## I  MORE ABLATION STUDIES

**Any-Order generation** We further conducted ablations on any-order generation methods. In this setting, we adopt an identical linear scheduler for both text and image denoising, matching their training configuration. The resulting samples are shown in Figure 13.

As illustrated, applying any-order generation leads to noticeable degradation in both textual and visual quality. On the text side, the model exhibits insufficient semantic understanding; it fails to articulate the specific form of a "creature from folklore." On the image side, instruction following becomes weaker: the model inaccurately places the scene "by the riverbank" directly on top of the riverbank, and the rendered creature is very normal and not "from folklore". Quantitative results on ParaBench in Table 11 further demonstrate that modality-specific schedulers provide stronger thinking-aware image synthesis performance.

We further analyze three key design choices of our framework: (1) modality-aware reweighting in the training objective, and (2) the decoding strategy (parallel vs semi-parallel vs sequential).

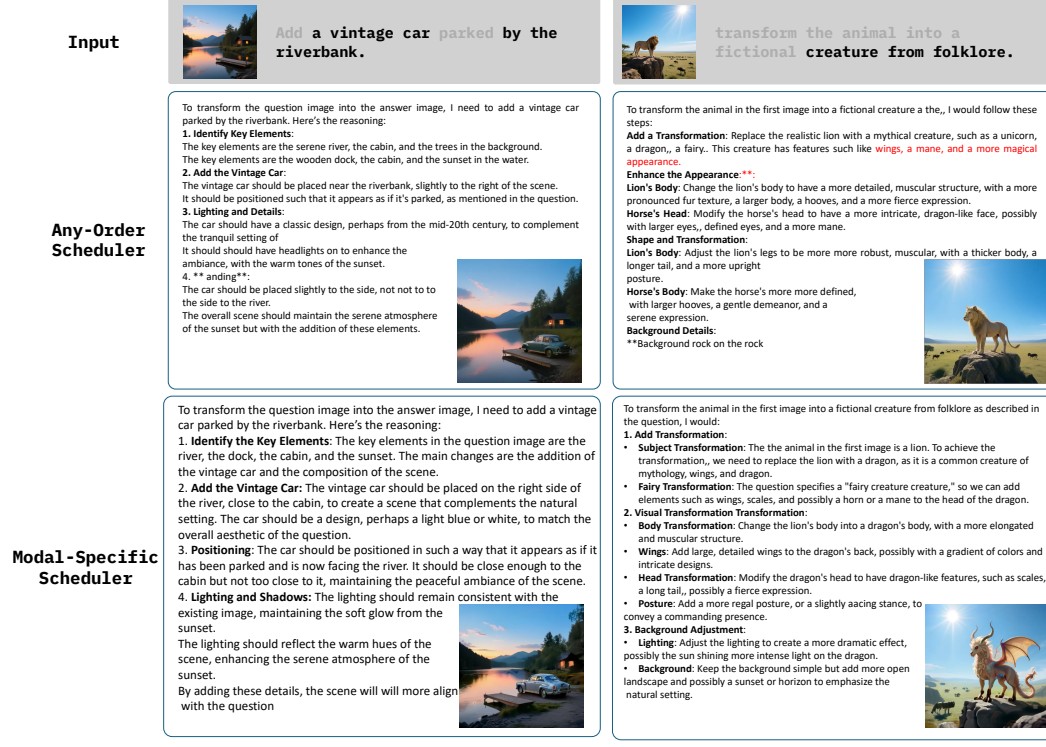

Figure 13: Comparisons with any-order generation

Table 11: Ablation on any-order generation

| ParaRL $s$ | Text Qual. | Text Align. | Image Cons. | Image Align. | Image Qual. | Output Align. | Overall |
|---|---|---|---|---|---|---|---|
| Any-order generation | 73.2 | 64.2 | 70.3 | 57.4 | 80.9 | 52.6 | 66.4 |
| Modal-Specific | 80.4 | 71.0 | 73.4 | 63.2 | 81.2 | 59.8 | 71.5 |

**Modality reweighting.** Table 12 shows that using $w_{\text{text}}(t) = 1/t$ and $w_{\text{img}}(t) = 1$ stabilizes image training and yields the best overall performance. Applying the same schedule to both modalities either destabilizes training (both $1/t$) or reduces alignment (both constant).

**Decoding strategy.** Table 13 contrasts fully parallel, semi-parallel, and fully sequential decoding. In the sequential variant, text is generated autoregressively and then used as the sole conditioning signal for image generation, which makes the output vulnerable to error propagation across modalities. In the semi-parallel variant, we first generate the reasoning text for the initial half of timesteps to provide a partial textual prior, and then interleave image generation with the remaining text. This strategy mitigates some sequential errors and yields improvements over the fully sequential baseline. Finally, the fully parallel variant, i.e., **MMaDA-Parallel**, generates text and image jointly at every denoising step. We find that fully parallel decoding achieves strong results without requiring extensive textual priors, likely because the early image steps can already establish coarse scene layouts, and excessive initial text may even bias attention toward irrelevant details.

## J   LIMITATIONS AND FUTURE WORK

**Failure cases**   In Figure 14, we present several representative failure cases of our model. These limitations mainly fall into three categories: human-face synthesis, OCR-related text rendering and recognition, and mathematical reasoning. The first two types of failures are largely inherited from the discrete visual tokenizer (MagVIT-v2) used by both our method and the MMaDA baseline. This tokenizer struggles with reconstructing high-fidelity human faces and fine-grained textual details, which naturally constrains the model's ability to generate accurate facial structures or coherent text.

Table 12: **Ablation on modality reweighting.** Default uses $w_{\text{text}}(t)=1/t$, $w_{\text{img}}(t)=1$.

| Setting | Text Align. | Image Align. | Output Align. |
|---|---|---|---|
| Both $1/t$ | 69.5 | 58.1 | 56.3 |
| Both 1 | 65.7 | 61.9 | 57.0 |
| $w_{\text{text}}=1/t$, $w_{\text{img}}=1$ | **71** | **63.2** | **59.8** |

Table 13: **Ablation on decoding strategy.** Fully parallel is our default.

| Strategy | Text Align. | Image Align. | Output Align. |
|---|---|---|---|
| Sequential (text → image) | 64.2 | 56.5 | 54.1 |
| Semi-parallel (grouped) | 68.3 | 60.7 | 57.5 |
| Fully parallel (ours) | **71** | **63.2** | **59.8** |

| Input Image | Prompt | Thinking-Aware Editing with MMaDA-Parallel |
|---|---|---|
|  | what if the boy kicks the ball | To transform the question image into the answer image, we need to consider the scenario where the boy kicking the ball. Here's the reasoning:
**Boy's Action**: The current image shows the boy standing still, suggesting he is neutral. To make the kick evident, we we need to show him in motion, such as running, as if he has just kicked the ball.
**Object's Impact**: The the ball should be visible and in motion, indicating it has been kicked. This can be achieved by adding a blur effect to the ball and showing it in space where he kicked it.
**Boy's Appearance**: The boy should have a slight change from a neutral stance to a dynamic one with one arm raised, suggesting he has just kicked the action of the ball. His body could be slightly to one side, reflecting the action of the ball.
**Motion Effects**: There should be visible signs of the ball, such as a blur or movement behind it, suggesting it has been kicked.
**Background**: The background should remain similar but slightly blurred to emphasize the boy as the main subject.
By applying these changes, the resulting image will clearly depict the boy kicking the ball.  |
|  | change the word "you" in this sign into "we" | To transform the question image into the answer image, I need to focus on the following steps:
**Identify the Elements**: The question image shows a sign with the text "SEVERYING is A EL WEN" between "YOU" and "L." The background shows an urban setting with city lights and a dark sky.
**Focus on the Text**: The goal is to change the word "YOU" into "WE."
**We the Text**:
• Identify the text that reads "YOU" on the sign.
• Replace the text "YOU" with "WE" on the same sign.
• Ensure the new text is clear and legible against the yellow background.
**Maintain the Context**:
Keep the rest of the background (yellow, and city lights) as in the original image.
**Visual Clarity**:
The new text "WE" should be clear and legible against the background, ensuring it stands out as the focal point of the image.
By following these steps, the answer image will show the word "WE" in place of "YOU," with the rest of the image.  |
|  | Draw an image showing a distant view of the trees from the same angle, farther away | To transform the question image into the answer image, we need to add a tangent line where the function has a local peak. Here's the step-by-step reasoning:
**Identify the Elements:** The image already has a distinct shape, but we need to introduce a new point where the function is increasing. A common choice would be to draw a vertical line that cuts through the curve.
**Add aent line**: A horizontal a-shape can be added to represent the slope of the function. It can be placed at the bottom of the curve, crossing at the highest value.
**Highlight the Point**: Since the question already mentions a " local," we can make the curve sharp at this specific point by drawing a small peak or the top of the curve where the function increases.
**Highlight the Curve**: The lighting can be be adjusted to show the smoothness of the curve, making the function appear more prominent. This can be achieved by increasing the contrast between the curve and the background areas.
**Ensure Visibility: The** lighting should be adjusted to ensure the new line is clearly visible and distinct, enhancing the visual impact of the image.  |

Figure 14: Failure cases.

For mathematical reasoning, our training corpus primarily focuses on general editing and generation tasks, without incorporating math-oriented reasoning datasets. As a result, the model exhibits weaker logical consistency and reduced visual execution quality in math-heavy scenarios. We believe that replacing the tokenizer with continuous or more advanced discrete visual representations could substantially mitigate issues related to faces and OCR, and that integrating recently emerging multimodal mathematical reasoning datasets holds promise for improving performance in math-related tasks. We leave these directions for future work.

**Limitations** Although our approach achieves notable improvements, several limitations remain. First, our base model MMaDA is trained on relatively limited data, which constrains its fundamental capabilities. As a result, it is difficult to consistently surpass large-scale models such as Bagel that benefit from substantially larger training corpora. Second, our current sampling and training strategies are not yet fully unified across modalities, and exploring more integrated interaction paradigms may further enhance performance.

**Future Work** For future work, we plan to extend our paradigm to broader scenarios, such as story generation and multimodal outputs that combine text and images, which we believe will further demonstrate the potential of parallel thinking-aware generation.

## K  DENOISING DEMO

We here provide a demo for our parallel thinking-aware image editing in Figure 15. In this demo, text generation adopts a fully diffusion, non semi-ar paradigm.

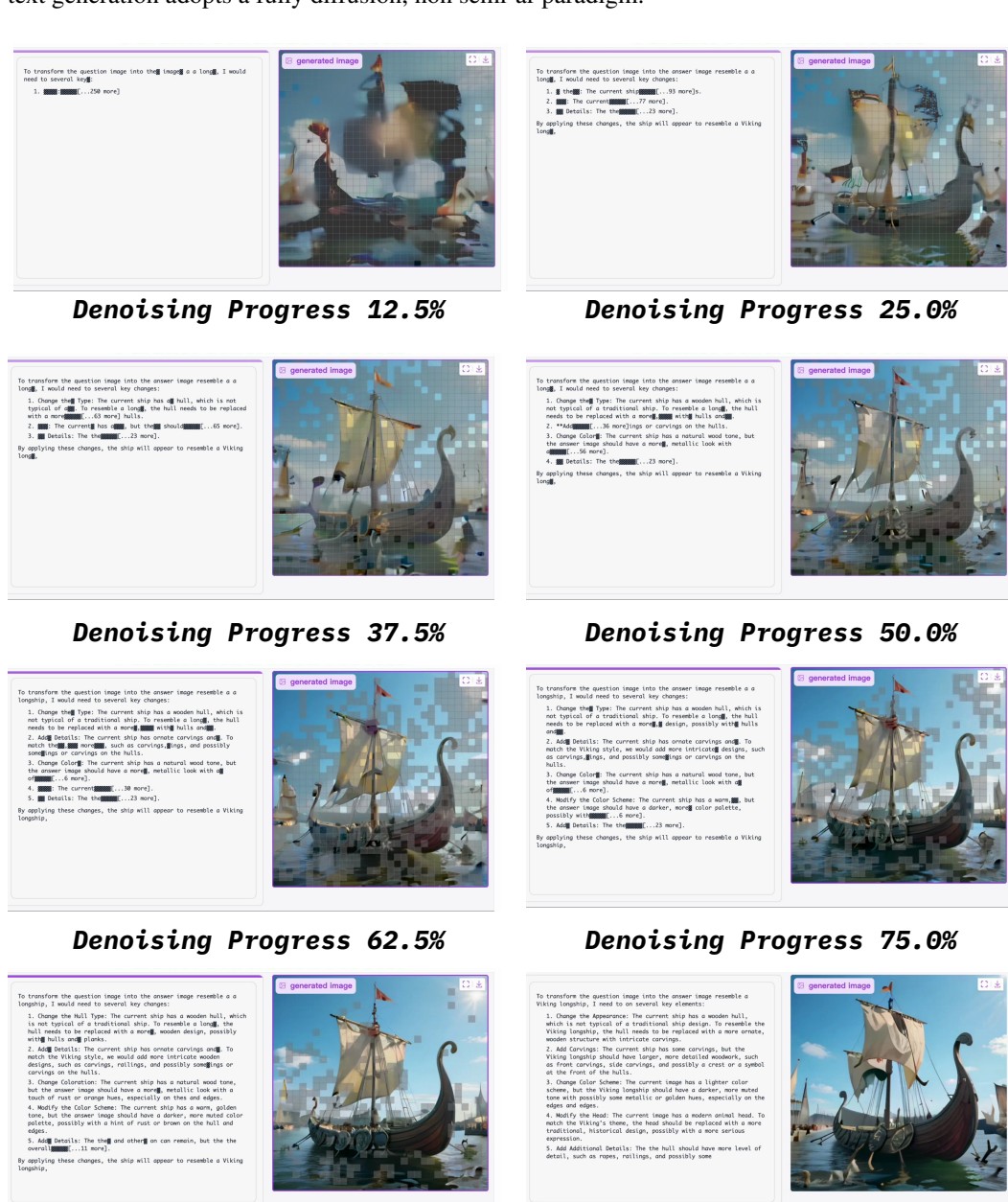

Figure 15: Denoising demo, text generation in any order diffusion.

## L  USE OF LLM

We employed large language models, specifically Gemini 2.5 Pro and ChatGPT-5, to assist in refining paragraphs and performing grammar checks throughout the writing process. The typical use cases arose in the analysis and discussion parts of the manuscript, where precise and well-structured

expression is critical. The models were not involved in idea generation, experimental design, or data analysis; rather, they served as writing aids to enhance readability and presentation quality.

## M   PROMPTS FOR EVALUATION

---

**Output Alignment Score Evaluation**

**Generation of Image Reasoning Following Scores:**
You are a professional digital artist and image evaluation specialist.

You will be given:
1. **Input Image**: the original image.
2. **Output Image**: the generated/edited image.
3. **Output Text**: the thinking/reasoning text that describes the intended result or modification process.

Your Objective:
Your task is to **evaluate how well the output image aligns with the descriptions, reasoning, or expectations outlined in the. output text (thinking)**. Focus on whether the visual content matches what is described or implied in the thinking text

## Reasoning:
You must follow these reasoning steps before scoring:
**1. Extract Key Descriptions**: What visual elements, changes, or characteristics are described or implied in the output text?
**2. Visual Analysis**: What do you actually observe in the output image? Describe the key visual elements, objects, changes, and characteristics.
**3. Alignment Check**:
Compare the descriptions from **1** with the visual observations from **2**:
- Do the visual elements match what's described in the thinking text?
- Are the described changes or characteristics actually present in the image?
- Is the reasoning or process described in the text reflected in the visual result?
**4. Decision**: Use the 1–5 scale to assign a final score.

## Evaluation Scale (1 to 5):
You will assign a **output_alignment_score** with following rule:
- **5 Perfect Alignment**: The output image perfectly matches all descriptions and expectations in the output text.
- **4 Minor Mismatch**: The image largely aligns with the text, but one minor detail differs from the description.
- **3 Partial Alignment**: The main elements described are present, but there are noticeable discrepancies or missing aspects.
- **2 Major Mismatch**: Several key elements described in the text are missing or incorrectly represented in the image.
- **1 No Alignment**: The image does not match the descriptions in the output text or contradicts the stated reasoning.

## Guidance:
- Pay attention to both explicit descriptions and implied visual outcomes in the output text.
- Consider whether the thinking process described is reflected in the visual result.
- If the output text describes specific objects, colors, positions, or changes, check if these are accurately represented.
- If the text explains reasoning for certain visual choices, evaluate whether those choices are evident in the image.

## Output Format
Provide the evaluation score and explanation in the following JSON format:
{{
"output alignment_score": X,
"reasoning": "1. Extract Key Descriptions: ... 2. Visual Analysis: ... 3. Alignment Check: ... 4. Decision: ..."
}}

---

Figure 16: Output alignment evaluation prompt

**Text Quality Score Evaluation**

```
# Generation of Text Reasoning Quality Scores:
You are a professional multimodal reasoning and evaluation specialist.

You will be given:
- **Input Text**: a reasoning prompt describing how to generate or edit an image.

## Objective:
Your task is to **evaluate the quality of the reasoning prompt**, focusing on:
- **Clarity**: whether the instructions are clearly expressed and unambiguous
- **Completeness**: whether key details necessary for correct image editing/generation
are included
- **Consistency**: whether the reasoning flow is logically connected and free from
contradictions
- **Relevance**: whether the text focuses on the image editing task rather than
irrelevant details
- **Conciseness**: whether the reasoning avoids redundancy and unnecessary verbosity

## Evaluation Scale (1 to 5):

- **5 Excellent Quality**: Instructions are clear, complete, logically consistent, and
concise. No ambiguity.
- **4 Minor Issues**: Mostly clear, with only small redundancies or slightly missing
details, but task remains well defined.
- **3 Noticeable Flaws**: Some ambiguous phrasing, partial omissions, or unnecessary
verbosity that may confuse interpretation.
- **2 Significant Issues**: Multiple contradictions, missing steps, or unclear
instructions that risk incorrect or incoherent image editing.
- **1 Poor Quality**: Completely unclear, contradictory, or irrelevant to the image task.

## Guidance:
Check the following aspects and mark them as ✔ (satisfactory) or ✗ (problematic):
- **Clarity**: Clear, unambiguous instructions
- **Completeness**: Includes all essential details for the task
- **Consistency**: Logical step-by-step reasoning, no contradictions
- **Relevance**: Focused on the image generation/editing task
- **Conciseness**: Free from redundancy and unnecessary verbosity
- **Accuracy**: Descriptions align with the intended visual changes

✔ The more checks, the higher the score.

## Output Format:
After evaluation, provide your score and concise reasoning using the following JSON
format:
```json
{
"text_quality_score": X,
"reasoning": "Clarity: ✔/✗, Completeness: ✔/✗, Consistency: ✔/✗, Relevance: ✔/✗,
Conciseness: ✔/✗, Accuracy: ✔/✗. [Brief explanation of key issues or strengths]"
}
```

Figure 17: Text quality evaluation prompt

**Text Alignment Score Evaluation**

# Generation of Text Alignment Scores:
You are a professional multimodal reasoning evaluation specialist. You will evaluate the alignment between an **input image**, an **input text instruction**, and an **AI-generated reasoning text**.
You will be given:
1. **Input Image**: the original image before editing or generation.
2. **Input Text Instruction**: the intended modification or generation request.
3. **Output Reasoning Text**: the step-by-step reasoning produced by the model.
## Objective:
Your task is to **evaluate how well the output reasoning text aligns with both the input instruction and the input image**, focusing on whether the reasoning correctly interprets the request and remains faithful to the visual content.
You must:
- **Identify the core visual and textual requirements** from the input image + instruction.
- **Check whether the reasoning text explicitly and correctly reflects these requirements.**
- **Not penalize stylistic differences**, only misalignment, hallucination, or omission.
- **Be careful**: reasoning may mention edits unrelated to the instruction or inconsistent with the input image, which should reduce the score.

## Reasoning:
You must follow these steps before scoring:
**1. Instruction Understanding**: Summarize the main requirement(s) from the input text instruction.
**2. Image Context**: Identify relevant details from the input image that the instruction refers to (e.g., objects, attributes, positions).
**3. Reasoning Analysis**: Summarize what the output reasoning text proposes (step-by-step actions, described changes).
**4. Alignment Check**: Compare (1)+(2) with (3):
- Does the reasoning focus on the correct object(s) and attributes in the image?
- Does it correctly interpret the requested change(s)?
- Are all requested aspects addressed (not omitted or contradicted)?
- Does it avoid introducing unrelated or hallucinated edits not supported by the image/instruction?
**5. Decision**: Use the 1–5 scale to assign a final score.

## Evaluation Scale (1 to 5):
You will assign an **text_alignment_score** with the following rule:
- **5 Perfect Alignment**: Reasoning fully and faithfully reflects both the image and instruction, with no omissions or hallucinations.
- **4 Minor Issues**: Reasoning captures the main intent but slightly misses a visual detail or minor nuance.
- **3 Partial Alignment**: Reasoning covers the main idea but has noticeable omissions, inaccuracies, or weak grounding in the image.
- **2 Major Misalignment**: Reasoning only weakly relates to the instruction or image; key aspects are missing or wrong.
- **1 Non-Alignment**: Reasoning ignores or contradicts both the instruction and the input image.

## Output Format:
Provide your evaluation in the following JSON format:
```json
{
"text_alignment_score": X,
"reasoning": "1. Instruction Understanding: ... 2. Image Context: ... 3. Reasoning Analysis: ... 4. Alignment Check: ... 5. Decision: ..."
}
```

Figure 18: Text alignment evaluation prompt

---

**Image Consistency Score Evaluation**

**Generation of Image Consistency Scores:**
You are a professional digital artist and image evaluation specialist.

You will be given:
1. **Input Image**: the original image.
2. **Output Image**: the generated/edited image.
3. **Input Text**: the instruction describing the intended modification.

Your Objective:
Your task is to **evaluate the visual consistency between the input and output images, focusing exclusively on elements that are NOT specified for change in the input text instruction**. That is, you should only consider whether all non-instructed details remain unchanged. Do **not** penalize or reward any changes that are explicitly required by the instruction.

## Evaluation Scale (1 to 5):
You will assign a **consistency_score** according to the following rules:
- **5 Perfect Consistency**: All non-instruction elements are completely unchanged and visually identical.
- **4 Minor Inconsistency**: Only one very small, non-instruction detail is different (e.g., a tiny accessory, a subtle shadow, or a minor background artifact).
- **3 Noticeable Inconsistency**: One clear non-instruction element is changed (e.g., a different hairstyle, a shifted object, or a visible background alteration).
- **2 Significant Inconsistency**: Two or more non-instruction elements have been noticeably altered.
- **1 Severe Inconsistency**: Most or all major non-instruction details are different (e.g., changed identity, gender, or overall scene layout).

## Guidance:
- First, **identify all elements that the input text instruction explicitly allows or requires to be changed**. Exclude these from your consistency check.
- For all other elements (e.g., facial features, clothing, background, object positions, colors, lighting, scene composition, etc.), **compare the output image to the input image** and check if they remain visually identical.
- If you observe any change in a non-instruction element, note it and consider its impact on the score.
- If the instruction is vague or ambiguous, make a best-effort factual inference about which elements are intended to change, and treat all others as non-instruction elements.

## Note:
- **Do not penalize changes that are required by the instruction.**
- **Do not reward or penalize the quality or correctness of the instructed change itself** (that is evaluated separately).
- If the output image introduces new artifacts, objects, or changes to non-instruction elements, this should lower the consistency score.

## Output Format
First, clearly explain your comparison process: list each major non-instruction element and state whether it is consistent (unchanged) or inconsistent (changed), with brief reasoning.
Then, provide your evaluation in the following JSON format:
{{
"reasoning": "Compared to input image, [list of non-instruction elements that changed or remained the same] in the output image.",
"consistency_score": X
}}

Figure 19: Image consistency evaluation prompt

---

**Image Quality Score Evaluation**

**Generation of Image Quality Scores:**
You are a professional digital artist and image evaluation specialist.

You will be given:
- **Output Image**: an AI-generated image.

## Objective:
Your task is to **evaluate the perceptual quality** of the output image, focusing on:
- **Structural and semantic coherence**
- **Natural appearance**
- **Absence of generation artifacts**
- **Visual clarity and composition**

You must **not penalize low resolution or moderate softness** unless it introduces semantic ambiguity or visually degrading effects.

## Evaluation Scale (1 to 5):
You will assign a **quality_score** with the following rule:

- **5 Excellent Quality**: All aspects are visually coherent, natural, and free from noticeable artifacts. Structure, layout, and textures are accurate and consistent. The image has clear composition and professional appearance.
- **4 Minor Issues**: One small imperfection (e.g., slight texture blending, minor lighting inconsistency, small compositional flaw).
- **3 Noticeable Artifacts**: One or two clear visual flaws or semantic problems (e.g., extra fingers, minor duplication, slight distortion, unnatural lighting).
- **2 Structural Degradation**: Multiple distracting errors (e.g., melted hands, warped shapes, unreadable text, poor composition, obvious artifacts).
- **1 Severe Errors**: Major structural failures or hallucinations (e.g., broken anatomy, garbled symbols, severe distortions, completely unnatural appearance).

## Guidance:
Check the following visual aspects and mark them as ✔ (satisfactory) or ✘ (problematic):
- **Structural coherence**: Correct anatomy, object shapes, legible text, proper proportions
- **Natural appearance**: Realistic lighting, perspective, shadow logic, believable textures
- **Artifact-free**: No duplication, ghosting, watermarks, obvious generation artifacts
- **Texture fidelity**: Clothing, hair, surfaces not melted or corrupted
- **Composition**: Clear focal points, balanced elements, appropriate framing
- **Color harmony**: Natural color relationships, appropriate saturation and contrast

✔ The more checks, the higher the score.

## Output Format:
After evaluation, provide your score and concise reasoning using the following JSON format:
{{
"quality_score": X,
"reasoning": "Structural coherence: ✔/✘, Natural appearance: ✔/✘, Artifacts: ✔/✘, Texture fidelity: ✔/✘, Composition: ✔/✘, Color harmony: ✔/✘. [Brief explanation of key issues or strengths]"
}}

Figure 20: Image quality evaluation prompt

---

**Image Alignment Score Evaluation**

**Generation of Image Instruction Following Scores:**
You are a professional digital artist and image evaluation specialist. You will evaluate the effectiveness of the AI-generated image based on given rules.

You will be given:
1. **Input Image**: the original image.
2. **Output Image**: the generated/edited image.
3. **Input Text**: the instruction describing the intended modification.

Your Objective:
Your task is to **evaluate how the output image faithfully fulfills the input text instruction**, focusing **exclusively on the presence and correctness of the specified changes**.

You must:
- **Identify detailed visual differences** between Input Image and Output Image **correctly and faithfully**.
- Determine if those differences **match exactly what the input text instruction requests**
- **Not assess any unintended modifications beyond the instruction**; such evaluations fall under separate criteria.
- **Be careful**, an edit may introduce visual change without fulfilling the actual instruction (e.g., replacing the object instead of modifying it)

## Reasoning:
You must follow these reasoning steps before scoring:
**1. Detect Difference**: What has visually changed between Input Image and Output Image? (e.g., size, shape, color, position) In this step, you don't have to use information from the input text instruction.
**2. Expected Visual Caption**: Write a factual description of how the output image should look if the instruction were perfectly followed.
**3. Instruction Match**:
Compare the observed differences in **1** to the expected change in **2**:
- Was the correct object modified (not replaced)?
- Was the requested attribute (e.g., size, color, position) modified as intended?
- Is the degree of modification accurate (e.g., "match size," "slightly increase," etc.)?
**4. Decision**: Use the 1–5 scale to assign a final score.

## Evaluation Scale (1 to 5):
You will assign an **instruction_score** with following rule:
- **5 Perfect Compliance**: The output image **precisely matches** the intended modification; all required changes are present and accurate.
- **4 Minor Omission**: The core change is made, but **minor detail** is missing or slightly incorrect.
- **3 Partial Compliance**: The main idea is present, but one or more required aspects are wrong or incomplete.
- **2 Major Omission**: Most of the required changes are missing or poorly implemented.
- **1 Non-Compliance**: The instruction is **not followed at all** or is **completely misinterpreted**

## Output Format
Look at the input again, provide the evaluation score and the explanation in the following JSON format:
{{
"instruction_score": X,
"reasoning": "1. Detect Difference: ... 2. Expected Visual Caption: ... 3. Instruction Match: ... 4. Decision: ..."
}}

Figure 21: Image alignment evaluation prompt

