# OpenReview forum: "MMaDA-Parallel: Multimodal Large Diffusion Language Models for Thinking-Aware Editing and Generation"
_ICLR.cc/2026/Conference — ICLR 2026 Poster_

### Official Review · Reviewer_w4zL · 2025-10-25

**Soundness:** 3
**Presentation:** 3
**Contribution:** 3
**Rating:** 6
**Confidence:** 4

**Summary:**

This paper addresses a critical and often overlooked failure mode in ​thinking-aware image synthesis, where the incorporation of a Chain-of-Thought (CoT) reasoning step can paradoxically degrade the final output due to error propagation in sequential, autoregressive pipelines. The authors systematically identify that this degradation is strongly correlated with poor alignment between the generated reasoning text and the final image.

To tackle this fundamental issue, the paper introduces a novel ​parallel multimodal diffusion framework​ that enables continuous, bidirectional interaction between text and images throughout the entire denoising trajectory. This paradigm shift from sequential to parallel generation allows for real-time grounding of textual descriptions in visual evidence and vice versa, effectively mitigating the accumulation of errors.

**Strengths:**

1. The paper introduces a paradigm shift by proposing a parallel multimodal diffusion framework to replace sequential autoregressive pipelines for thinking-aware generation. This creative combination of discrete diffusion with bidirectional cross-modal attention effectively addresses the critical failure mode of error propagation identified in existing models.

2. The proposed ParaRL algorithm, which applies reinforcement learning along the denoising trajectory, further represents a significant conceptual advance in optimizing for intermediate alignment.

3. The paper is exceptionally well-written and structured, with a clear narrative that logically progresses from problem identification to solution and validation.

4. It establishes a more robust paradigm for thinking-aware image synthesis, with potential broad impact on the field of multimodal generation. The introduction of the ParaBench benchmark fills a critical gap by providing a diagnostic tool to evaluate cross-modal alignment, which will be important.

**Weaknesses:**

Please see my questions.

**Questions:**

1. In Figure 3, the denoising process for text is depicted as sequential sentence-by-sentence generation. However, in inference, the denoising of text spans should occur in an arbitrary, non-sequential order. Could the authors clarify this, otherwise the diagram and method might cause confusion.

2. In the Appendix experiments, which model does "MMaDA-sequential" refer to? Is it the "MMaDA-MixCoT"? It appears that "MMaDA-parallel" only slightly outperforms "MMaDA-sequential." If "MMaDA-sequential" is fine-tuned using the proposed constructed data, would there be further performance improvements?

3. The generated CoT text relies on "Qwen2.5-VL-7B." How is the high quality of the constructed text ensured? Why wasn't a larger model, such as a 72B version, used for labeling? If higher-quality reasoning texts were constructed, could this further enhance performance?

4. In parallel reinforcement learning, is using "Qwen-VL" to evaluate the semantic alignment between partial text and partially generated images reliable? Since intermediate generated images might still be blurry or meaningless, the VLM's preference may not effectively judge meaningful scores.

5. Although MMaDA-Parallel achieves higher output alignment on ParaBench compared to other open-source models like Bagel, its output image quality does not surpass them. In other words, how can the author prove that output alignment is positively correlated with generating high-quality target images?

---

> ### Author Response · Authors · 2025-11-24
> **Response to Reviewer w4zL (Part 1/2)**
>
> We sincerely thank the reviewer for your exceptionally thoughtful and encouraging evaluation. We deeply appreciate your recognition of the paradigm shift introduced by our parallel multimodal diffusion framework, the conceptual advance of trajectory-level optimization in ParaRL, the clarity of our writing, and the importance of ParaBench as a diagnostic tool for cross-modal alignment. We address your concerns in detail as follows:
>
> **Q1: Clarification of Figure 3 and text denoising order**
>
> Your understanding is completely accurate. The behavior shown in Figure 3 originates from the “semi-autoregressive” (semi-AR) text sampling strategy used by LLaDA and MMaDA, as we explained in **Appendix E (“Text Token Denoising”)**. This is the default inference method adopted by these unified diffusion LLMs, and since our model is a direct adaptation of MMaDA, we keep this inference mode unchanged.
> Under semi-AR sampling, the model denoises from block to block, but within each block, token updates may occur at arbitrary positions. This sometimes leads to slightly more stable or coherent early-stage text, which is why the visualization may appear sequential.
>
> Importantly, as you correctly point out, this behavior is not fixed. Following your suggestion, we tested **fully non-sequential denoising (without semi-AR sampling)**, and found that both the generated text and the final images remain high-quality, shown in Table a. We include a teaser in **Page 28, Appendix K Figure 15**, showing the step-by-step evolution of text and images, where it is clear that text tokens are refined at arbitrary locations across the trajectory.
>
> Table a. Comparisons of semi-ar decoding and fully non-sequential denoising
> | Decoding Strategy             | Text Qual. | Text Align. | Image Cons. | Image Align. | Image Qual. | Output Align. | Overall |
> |------------------------------|------------|-------------|-------------|--------------|-------------|----------------|---------|
> | ully non-sequential denoising          | 79.2       | 69.1        | 73.1        | 62.7         | 80.5        | 58.6           | 70.5   |
> | Semi-autoregressive | **80.4**   | **71.0**    | **73.4**    | **63.2**     | **81.2**    | **59.8**       | **71.5** |
>
>
> **Q2: Clarification about “MMaDA-sequential”**
>
> Thank you for the question. We apologize for not explaining this more clearly. “MMaDA-sequential” refers to our fine-tuned sequential baseline, not MMaDA-MixCoT. It is trained on the same constructed data and with the same tuning recipe as MMaDA-Parallel to ensure a fair comparison. We have updated the **Appendix B.2** to make this clearer.
>
>
> **Q3: Using a stronger LLM for CoT curation**
>
> Thank you for the constructive question. We did observe in preliminary experiments that Qwen2.5-VL-72B produces slightly higher-quality reasoning traces. However, since our CoT sequences are short (≈256 tokens), we found that the difference between 7B and 72B is limited for most cases. We therefore used the 7B model to reduce the cost of large-scale labeling.
>
> Following your suggestion, we additionally generated CoT traces with a stronger model. Due to resource limits (Qwen-VL-72B would require >2 weeks to run locally, and we do not have API access), we used GPT-4.1 to generate reasoning traces. The ParaBench results are:
>
> Table b. Ablations of stronger LLM for CoT generation.
> | CoT Source           | Text Qual. | Text Align. | Image Cons. | Image Align. | Image Qual. | Output Align. | Overall |
> |------------------------------|------------|-------------|-------------|--------------|-------------|----------------|---------|
> | Qwen2.5-VL-7B          |  80.4   | 71.0    | 73.4    | 63.2     | 81.2    | 59.8       | 71.5 |
> | GPT-4.1 | 82.1   | 71.3    | 73.2    | 62.8     | 82.1    | 59.5       | **71.8** |
>
> We do observe that higher-quality CoT traces yield better textual reasoning, which further improves the performance of thinking-aware image synthesis. We sincerely thank the reviewer for this valuable suggestion and will include the ablation for Qwen2.5-VL-72B in the final version of the paper.
>
> Although there is improvement, the gain is relatively small compared with the additional computational cost. This is likely because our current tasks do not require long or deeply structured reasoning. In future extensions involving more complex scenarios, such as compositional generation or region-level editing, where longer and more precise reasoning chains are essential, we expect stronger benefits from enhancing the quality of CoT traces.

---

> ### Author Response · Authors · 2025-11-24
> **Response to Reviewer w4zL (Part 2/2)**
>
> **Q4:  Reliability of using VLM rewards for partial images**
>
> Thank you for this accurate observation. Firstly, we actually use CLIP rather than Qwen-VL as the reward model. Secondly, regarding your concerns about this reliability, we find that this intermediate blur does not significantly affect reward quality. These blurs mainly occur at early steps of denoising, and we compute rewards on the decoded x0 images. Even though the decoded images are incomplete, they already contain enough coarse structure, such as **objects, layout, and major attributes**, for CLIP to produce reliable alignment signals with the text. You can also refer to our demo in **Appendix K Figure 15**. As we only compute $s=3$ steps in ParaRL, the intermediate results from these steps can yield meaningful scores for computing group-relative reward advantages.
>
> **Q5: Alignment improves image quality**
> Thank you for this valuable question. We address your concern from three perspectives:
> 1. **Comparison with our sequential variant.**
> As you noted in Q2, MMaDA-Parallel consistently produces higher image quality and stronger instruction following than the sequential variant under identical conditions (Appendix B). We also report both models’ ParaBench metrics below:
>
>     Table b. Comparisons of ParaBench between MMaDA-Parallel and MMaDA-Sequential
>     | **Method**           | Text Qual. | Text Align. | Image Cons. | Image Align. | Image Qual. | Output Align. | Overall         |
>     | -------------------- | ---------- | ----------- | ----------- | ------------ | ----------- | ------------- | -------- |
>     | **MMaDA-Sequential** | 76.2       | 66.8        | 69.4        | 55.7         | **78.1**        | 63.4          |  68.3        |
>     | **MMaDA-Parallel**    | 80.4     | 71.0     | *73.4*      | 63.2     | **81.2**      | 59.8      | **71.5**        |
>
>
>    This controlled comparison suggests that better alignment directly correlates with more coherent and higher-quality images, as the main training differences in these two variants are the mutual attention between output modalities and our reward for output alignment enhancement.
>
> 2. **Why alignment contributes to image quality.**
> Many image-quality failures arise from misunderstanding world knowledge or missing fine-grained reasoning details. When alignment improves, the model correctly understands scene relations, object attributes, and contextual constraints, which directly enhances visual accuracy. For example, better alignment helps the model infer **correct spatial relations, detailed attributes, and fine-grained semantics** that the image must reflect. This naturally results in clearer, more coherent, and more faithful images, mostly with greater quality.
>
>     Thus, stronger alignment ensures that reasoning details are correctly translated into visual features, which improves image quality even without explicitly tuning for perceptual metrics.
>
> 3. **Scaling the data.**
> Absolute image quality often depends more on the underlying model capacity, tokenizer fidelity, and data quality rather than on alignment alone. Our main contribution is the parallel generation framework, not large-scale data engineering or heavy optimization. The original MMaDA tokenizer imposes clear limitations on visual fidelity, which affect both sequential and parallel variants.
> To isolate this bottleneck, we further applied our parallel framework to Lumina DiMOO[1], which uses the much stronger amused-VQ tokenizer and is trained on larger and higher-quality datasets. We name this variant MMaDA-Parallel-A. After applying parallel generation and ParaRL, MMaDA-Parallel-A achieves improved alignment and **also improved image quality**, surpassing Bagel in several ParaBench categories. Below, we provide a comparison of this variant, where sequential and parallel training are the only differences. This shows that once the backbone and tokenizer are strong enough, alignment gains do translate into higher image quality.
>
>     Table c. Scaling the data, trained from Lumina-DiMOO.
>     | Method                 | Text Qual. | Text Align. | Image Cons. | Image Align. | Image Qual. | Output Align. | Overall |
>     |------------------------|------------|-------------|-------------|--------------|-------------|----------------|---------|
>     | **MMaDA-Parallel-A**   | **84.1**   | **76.5**    | 71.0        | **67.8**     | **83.6**    | **68.8**       | **75.3** |
>     | **MMaDA-Sequential-A** | 83.6       | 75.2        | 71.3        | 65.9         | 84.2        | 65.6           | 74.3  |
>
> We additionally include its quantitative and qualitative results in **Page 16, Appendix A, Figure 8 and Table 6**, which show consistent improvements in alignment and reasoning quality.
>
>
> [1] Xin, Yi, et al. "Lumina-dimoo: An omni diffusion large language model for multi-modal generation and understanding." arXiv preprint arXiv:2510.06308 (2025).

---

> > ### Comment · Reviewer_w4zL · 2025-11-28
> > **Reply to the authors**
> >
> > I thank the authors for their detailed rebuttal, which has successfully addressed all of my concerns. I have no further questions at this time and will maintain my positive rating.

---

> > > ### Author Response · Authors · 2025-11-28
> > >
> > > Dear Reviewer w4zL:
> > >
> > > We are very pleased that our rebuttal has successfully addressed your concerns. Your feedback has been invaluable for our research.
> > >
> > > Warm regards,
> > >
> > > The Authors

---

### Official Review · Reviewer_nFAx · 2025-10-28

**Soundness:** 3
**Presentation:** 3
**Contribution:** 2
**Rating:** 4
**Confidence:** 4

**Summary:**

This paper investigates the failure modes of thinking-aware generation and identifies that sequential reasoning–generation pipelines often suffer from error propagation. To systematically analyze this issue, the authors introduce ParaBench, a new benchmark that jointly evaluates reasoning and image outputs. Building upon this insight, they propose MMaDA-Parallel, a diffusion-based framework enabling continuous text–image interaction, and further optimize it via Parallel Reinforcement Learning (ParaRL). Experiments on ParaBench show a 6.9% improvement in output alignment over Bagel, demonstrating enhanced cross-modal consistency.

**Strengths:**

1.The paper provides a clear and insightful analysis of failure modes in thinking-aware multimodal generation, highlighting a real problem in current reasoning–generation pipelines.

2.ParaBench offers a valuable evaluation framework that jointly measures reasoning and image alignment, which could be useful for future multimodal research.

3.The proposed parallel diffusion framework with stepwise semantic optimization is well-motivated and achieves measurable gains in cross-modal alignment.

**Weaknesses:**

1. The method novelty is limited — the proposed MMaDA-Parallel and ParaRL mainly combine existing ideas of diffusion fine-tuning and reinforcement learning under a parallel setting.

2.The reported 6.9% improvement in output alignment, while positive, appears modest given the additional complexity and training cost. The authors are encouraged to provide stronger evidence that this gain is statistically or practically significant, or to further improve the results through more comprehensive experiments.

**Questions:**

see the weakness

---

> ### Author Response · Authors · 2025-11-24
> **Response to Reviewer nFAx (Part 1/3)**
>
> We thank the reviewer for acknowledging our "insightful" analysis, evaluation framework, and parallel generation design. We would like to clarify that the core innovation of our work lies in proposing a **comprehensive and scalable parallel generation framework**, rather than introducing only a single method. The framework is **model-agnostic**, and its improvements **consistently transfer across different baselines**, demonstrating robustness beyond any specific base model.
>
> Here are our detailed responses.
>
> **Q1: Limitations of novelty of MMaDA-Parallel and ParaRL**
>
> Our work is not designed merely to present methods. Instead, our intention is to introduce a **comprehensive and unified new framework** that covers a new scenario, a new benchmark, a new model architecture, and a new RL algorithm, providing a new pathway for future research in thinking-aware image synthesis.  Notably, other reviewers have explicitly recognized the novelty of our contributions, describing them as “new model design” (Reviewer rn82), “insightful motivations” (Reviewer WaTC), and a “creative combination” (Reviewer w4zl). We summarize our core innovations as follows:
>
> 1. **Novel scenario and evaluation perspective**: First, one of our core contributions is our evaluation benchmark, ParaBench. Our work begins with **a new empirical observation**: sequential thinking-aware image synthesis can degrade image quality due to a misalignment between reasoning and generation. This failure mode was not analyzed or quantified in prior multimodal diffusion research. To address this missing perspective, we propose ParaBench. To the best of our knowledge, this is the **first benchmark** to jointly evaluate generated reasoning and image outputs with **explicit output alignment metrics**. It is also a general benchmark for thinking-aware **image editing and generation**. The design of ParaBench is novel and not shown before, and "with potential broad impact on the field of multimodal generation. " (Acknowledgment by Reviewer w4zL)
>
> 2. **New formulation of unified discrete diffusion models**:
> While prior works such as Show-o[1], Janus[2], Bagel[3], and MMaDA[4] adopt unified multimodal architectures, their unification is limited to **parameter sharing** and **multi-task capability.** These models can perform image understanding or image generation, but each task must be executed **one at a time**. In other words, the “unified” design in earlier systems does not enable joint or concurrent multimodal generation; the model only switches between tasks within a shared backbone.
>
>     In contrast, our proposed MMaDA-Parallel provides a **new formulation** of unified diffusion models. We extend the unified discrete diffusion architecture to support  **parallel multimodal generation**,  which autoregressive pipelines cannot achieve due to their inherently sequential nature. This formulation leverages the property of diffusion models that allow multiple modalities to be denoised concurrently within a shared forward process. To the best of our knowledge, this work is **the first** to demonstrate that unified discrete diffusion models can be naturally extended beyond multi-task usage into the domain of parallel multimodal generation, offering **"a new model design"** for thinking-aware image editing and generation. (Acknowledgment by Reviewer rn82)
>
> 3. **Methodological novelty in ParaRL: trajectory-level semantic optimization**. ParaRL is not a simple integration of diffusion fine-tuning and RL. Prior diffusion RL approaches predominantly optimize output **outcome level** rewards, as seen in Flow-GRPO[5], DanceGRPO[6], and DDPO[7], and they do not support **trajectory-level rewards.** This is because traditional diffusion models provide no reliable way to assign rewards at intermediate denoising steps. In contrast, our parallel generation paradigm **naturally offers multimodal alignment signals** at each diffusion step, since the text and image trajectories evolve jointly. This enables us to compute stepwise multimodal semantic rewards throughout the denoising trajectory, rather than only at the final output. This capability **is unique to our setting** and has not been explored in prior work. It forms a key methodological innovation of ParaRL, and "represents a significant conceptual advance". (Reviewer w4zL)
>
> In summary, the combination of ParaBench, MMaDA-Parallel, and ParaRL provides **a new perspective** for multimodal diffusion generation—introducing new evaluation tools, new generation paradigms, and new optimization mechanisms. We believe these contributions will be valuable for future research on unified multimodal diffusion models and are far from a simple combination of existing techniques.
>
> We hope this clarification helps convey the methodological and conceptual novelty of our work more clearly, and we would kindly ask the reviewer to reconsider your assessment of our contribution and originality.

---

> ### Author Response · Authors · 2025-11-24
> **Response to Reviewer nFAx (Part 2/3)**
>
> **Q2: Concern about whether the 6.9% improvement is significant given complexity and training cost.**
>
> We thank the reviewer for the constructive suggestion. We clarify the significance of the improvement from both the *cost–performance* perspective and from *additional experiments* performed on stronger baselines.
>
> 1. **This improvement is not modest, and our training cost and complexity are actually small**
>
> The key goal of our work is to demonstrate that the parallel paradigm outperforms sequential thinking-aware image synthesis. We do not aim to achieve full SOTA performance under massive training budgets, because our available computational resources are very limited. Below is a comparison among Bagel[3], our base model (MMaDA-8B), and MMaDA-Parallel:
>
> Table a. Comparisons of training cost.
> | Model                     | Training Data | Compute Resources (80G GPU) |
> | ------------------------- | ------------- | ---------------------------- |
> | **Bagel**                 | **2665M**     | **>1000 GPUs**               |
> | **MMaDA-8B**              | ~30M          | ~64 GPUs                     |
> | **MMaDA-Parallel (Ours)** | **100K**      | **16 GPUs**                  |
>
> For complexity, under standard settings, introducing parallel generation requires producing 256 text tokens + 1024 image tokens simultaneously. In editing tasks, the transformer receives both modalities as input (256 text + 1024 image), and the original version of MMaDA only outputs 1024 image tokens. The relative complexity can be approximated as: $\frac{(1024 + 256 + 1024 + 256)^2}{(1024 + 256 + 1024)^2} \approx 1.23$
>
> This means that parallel generation increases computational complexity by only ~23%, and given the above massive gap between data and resources, it is neither reasonable nor expected for MMaDA-Parallel to surpass Bagel universally. Yet, with only 100K data and 16 GPUs, our method achieves a 6.9% improvement in output alignment and, in several categories, surpasses Bagel itself. This is a **substantial** and meaningful improvement at an extremely low cost and minimal complexity overhead.
>
> When comparing with Show-o (tuned), our method achieves a 4.9% higher overall ParaBench score under the same fine-tuning data and training cost. This comparison is particularly fair and meaningful because Show-o is a unified model that generates text autoregressively and images via discrete diffusion, and its overall training budget is much closer to MMaDA than Bagel's. Therefore, this improvement directly reflects the benefit of our **parallel diffusion design**, further validating the effectiveness of the proposed paradigm.

---

> ### Author Response · Authors · 2025-11-24
> **Response to Reviewer nFAx (Part 3/3)**
>
> **Q2: Concern about whether the 6.9% improvement is significant given complexity and training cost.**
>
> We thank the reviewer for the constructive suggestion. We clarify the significance of the improvement from both the *cost–performance* perspective and from *additional experiments* performed on stronger baselines.
>
> 2.**Additional experiments**
>
> Our core innovation lies in introducing this new scenario and establishing a unified parallel framework for thinking-aware image synthesis. We do not focus on extensive performance tuning, and surpassing Bagel through large-scale data engineering or heavy optimization is not the primary goal of our work. In fact, the performance of our model is strongly constrained by the visual tokenizer used in the original MMaDA pipeline, which introduces bottlenecks for visual fidelity and alignment.
>
> To address the reviewer’s suggestion, we conducted additional experiments by replacing the visual tokenizer with the amused-VQ tokenizer and began our parallel adaptation on Lumina DiMOO[8], a recent discrete diffusion framework that uses this improved tokenizer and is trained on larger and higher-quality datasets. The results on ParaBench are shown below:
>
>  Table b. Additional comparisons of quantitative results on Parabench, with Lumina-DiMOO as the base model.
>  | **Method**                                   | Text Qual. |Text Align.| Image Cons. |Image Align.| Image Qual.| Output Align.| Overall|
> | -------------------------------------------- | -------- | -------- | ----------- | -------- | --------- | --------- | ----------- |
> | **Bagel (w/ think)**                         | 82.0     | *74.5*   | **76.7**    | 63.4     | 81.5      | 52.9      | 71.8        |
> | **Show-o** (tuned)                          | 75.2     | 70.7     | 69.1        | 57.5     | 78.5      | 48.9      | 66.6        |
> | MMaDA-Parallel (ours) w/o ParaRL        | 76.5     | 70.4     | 70.5        | 58.2     | 80.5      | 51.5      | 67.9        |
> | MMaDA-Parallel (ours) w/ ParaRL          | 80.4     | 71.0     | *73.4*      | 63.2     | 81.2      | 59.8      | 71.5        |
> | **MMaDA-Parallel-A** (ours) w/o ParaRL | *82.6*   | 73.7     | 71.3      | *64.6* | *82.6*    | *63.3*    | *73.0*      |
> | **MMaDA-Parallel-A** (ours) w/ ParaRL  | **84.1** | **76.5** | 71.0     | **67.8** | **83.6**  | **68.8**  | **75.3**    |
>
> These results demonstrate that with a stronger tokenizer and backbone, our parallel paradigm produces substantial improvements across almost all categories, in many cases surpassing Bagel. This further confirms that the parallel paradigm provides **practical, robust, and scalable** benefits. We additionally include its quantitative and qualitative results in **Page 16, Appendix A, Figure 8 and Table 6**, which show consistent improvements in alignment and reasoning quality. We hope these results can further validate our innovations and address your concerns.
>
>
> [1] Wu, Chengyue, et al. "Janus: Decoupling visual encoding for unified multimodal understanding and generation." Proceedings of the Computer Vision and Pattern Recognition Conference. 2025.
>
> [2] Xie, Jinheng, et al. "Show-o: One Single Transformer to Unify Multimodal Understanding and Generation." The Thirteenth International Conference on Learning Representations.
>
> [3] Deng, Chaorui, et al. "Emerging properties in unified multimodal pretraining." arXiv preprint arXiv:2505.14683 (2025).
>
> [4] Yang, Ling, et al. "Mmada: Multimodal large diffusion language models." arXiv preprint arXiv:2505.15809 (2025).
>
> [5]Liu, Jie, et al. "Flow-grpo: Training flow matching models via online rl." arXiv preprint arXiv:2505.05470 (2025).
>
> [6] Xue, Zeyue, et al. "DanceGRPO: Unleashing GRPO on Visual Generation." arXiv preprint arXiv:2505.07818 (2025).
>
> [7] Black, Kevin, et al. "Training diffusion models with reinforcement learning." arXiv preprint arXiv:2305.13301 (2023).
>
> [8] Xin, Yi, et al. "Lumina-dimoo: An omni diffusion large language model for multi-modal generation and understanding." arXiv preprint arXiv:2510.06308 (2025).

---

> > ### Comment · Reviewer_nFAx · 2025-11-28
> >
> > Firstly, I have read your rebuttal very carefully. Thanks for your response.
> >
> > I think your response have addressed my concern and  after reading other reviews, I also think this work is good.
> >
> > I decide to raise my score to 6 when I can.
> >
> > Best

---

> > > ### Author Response · Authors · 2025-11-28
> > >
> > > Dear Reviewer nFAx,
> > >
> > > We are very pleased to hear that our rebuttal has successfully addressed your concerns. Your constructive feedback has been truly valuable to our work, and we will make sure to incorporate the clarified innovations and the additional experimental results into the final version of the paper.
> > >
> > > Warm regards,
> > > The Authors

---

### Official Review · Reviewer_WaTC · 2025-11-01

**Soundness:** 3
**Presentation:** 3
**Contribution:** 3
**Rating:** 6
**Confidence:** 3

**Summary:**

This paper focuses on an important problem in image generation, that the inclusion of reasoning can reduce the semantic fidelity of the generated images. It identifies that this could be caused by the misalignment of the model’s generated reasoning and its final image. To solve this problem, the authors propose bidirectional attention between modalities at every denoising step, and optimize alignment along the denoising trajectory. Experimental results show that the proposed method achieves significant improvement over baseline methods.

**Strengths:**

1. The proposed ParaBench is an effective tool for the analysis of thinking-aware image synthesis.  The finding regarding the strong correlation between performance degradation and poor alignment between the generated modalities is interesting and insightful.
2. This paper explains the method design clearly, with insightful motivations.
3. Visualization results in Figure 5 are impressive, showing the improvement in challenging scenarios such as the compositional settings.
4. The paper is well-written and easy to follow.

**Weaknesses:**

1. This paper shows the improvements of the paper based on the proposed ParaBench (Table 2). How about standard public benchmarks used in the original Bagel paper, such as  GenEval[1], WISE[2], GEdit-Bench[3]?

[1]  Geneval: An object-focused framework for evaluating text-to-image alignment. NeurIPS, 2023.

[2] Wise: A world knowledge-informed semantic evaluation for text-to-image generation. arXiv preprint arXiv:2503.07265, 2025.

[3] Step1x-edit: A practical framework for general image editing. arXiv preprint arXiv:2504.17761, 2025.

2.  Figure 5 shows some results of the proposed method.  What is the failure case of the proposed method?

**Questions:**

Please refer to the weaknesses.

---

> ### Author Response · Authors · 2025-11-24
> **Response to Reviewer WaTC (Part 1/2)**
>
> We sincerely thank the reviewer for your acknowledgement of ParaBench's effectiveness in analyzing thinking-aware image synthesis, our findings being "interesting and insightful", our methods with "insightful motivations", and the visualization results being impressive. We address your concerns as follows:
>
> **Q1: Additional Standard Benchmark?**
>
> Thanks for your constructive suggestion. We now report these results as follows:
> 1. **GenEval**. Actually, we already reported these results in Appendix B.2, Table 7. For your convenience, we report it again here:
>
> Table A. Quantitative results on GenEval.
> | **Method**                | **Single Obj.** | **Two Obj.** | **Counting** | **Colors** | **Position** | **Color Attr.** | **Overall** |
> | ------------------------- | --------------- | ------------ | ------------ | ---------- | ------------ | --------------- | ----------- |
> | SDXL                      | **0.98**        | 0.74         | 0.39         | **0.85**   | 0.15         | 0.23            | 0.55        |
> | Show-o                    | 0.95            | 0.52         | 0.49         | 0.82       | 0.11         | 0.28            | 0.53        |
> | MMaDA                     | 0.99            | 0.76         | 0.61         | 0.84       | 0.20         | 0.37            | 0.63        |
> | Bagel                     | 0.98            | **0.95**     | **0.84**     | 0.95       | **0.78**     | **0.77**        | **0.88**    |
> | MMaDA (Sequential)        | 0.99            | 0.78         | 0.66         | 0.87       | 0.34         | 0.37            | 0.68        |
> | **MMaDA-Parallel (Ours)** | **0.99**        | 0.83         | 0.70         | 0.88       | 0.40         | 0.47            | 0.71        |
>
>
> 2. **WISE and GEdit-Bench**. Following your suggestion, we additionally evaluated our Sequential and Parallel variants of MMaDA on WISE and GEdit-Bench. Due to the large gap in training data between Bagel and our model, a direct comparison with Bagel is not meaningful. Thus, consistent with our GenEval reporting, we focus on the **relative comparison** between the Sequential and Parallel training paradigms under identical training cost.
>
> Table B. Quantitative results on WISE.
> | Model            | Cultural | Time | Space | Biology | Physics | Chemistry | Overall |
> |------------------|----------|------|-------|---------|---------|-----------|---------|
> | SDXL   | 0.43     | 0.48 | 0.47  | 0.44    | 0.45    | 0.27      | 0.43    |
> | show-o           | 0.28     | 0.36 | 0.40  | 0.23    | 0.33    | 0.22      | 0.30    |
> | Bagel            | 0.44     | 0.55 | 0.68  | 0.44    | 0.60    | 0.39      | 0.52    |
> | MMaDA-Sequential  | 0.39     | 0.54 | 0.58  | 0.55    | 0.44    | 0.22      | 0.44    |
> | MMaDA-Parallel    | 0.42     | 0.56 | 0.59  | 0.57    | 0.47    | 0.24      | 0.47    |
>
> Table C. Quantitative results on Gedit-Bench.
> | Model            | G_SC | G_PQ | G_O |
> |------------------|------|------|------|
> | Gemini 2.0       | 6.73 | 6.61 | 6.32 |
> | GPT-4o           | **7.85** | **7.62** | **7.53** |
> | Instruct-Pix2Pix | 3.58 | 5.49 | 3.68 |
> | MagicBrush       | 4.68 | 5.66 | 4.52 |
> | AnyEdit          | 3.18 | 5.82 | 3.21 |
> | Step1X-Edit      | 7.09 | 6.76 | 6.70 |
> |*BaGEL        | 7.36 | 6.83 | 6.52 |
> | MMaDA-Sequential | 5.63 | 5.97 | 5.13 |
> | **MMaDA-Parallel** | **5.72** | **6.28** | **5.23** |
>
> These additional results, also reported in our revised manuscript **Page 19 & 20, Appendix B, Table 8 & 9**, once again support our core claim: Strengthening the output alignment between reasoning and image synthesis directly enhances final image quality and instruction-following capabilities at the same training cost. Thank you again for your constructive suggestions.

---

> ### Author Response · Authors · 2025-11-24
> **Response to Reviewer WaTC (Part 2/2)**
>
> **Q2: Failure cases?**
>
> We have included several representative failure cases in **Page 27, Figure 14 of Appendix J.** The failure modes of our method mainly fall into three categories: (1) **Human faces**, (2) **OCR-related text generation and recognition**, (3) **Mathematical reasoning tasks.**
>
> For **(1) human faces** and **(2) OCR/text understanding**, we attribute the failures to the limitations of the **discrete visual tokenizer**. Similar to the base model MMaDA, our method relies on MagVIT-v2. Its reconstruction quality on realistic human faces and fine-grained text is still suboptimal. These tokenizer limitations directly restrict the model’s ability to generate high-fidelity faces or render coherent text. Thus, in these scenarios, making further improvements with our MMaDA-Parallel is more than challenging.
>
> For (3) **mathematical reasoning**, our training data focuses primarily on general editing and generation tasks. We did not incorporate math reasoning datasets, and therefore, the model lacks reasoning consistency, and visual execution quality degrades in this domain.
>
> For issues (1) and (2), we believe that adopting continuous visual tokenizers or more advanced discrete tokenizers could significantly alleviate these limitations. Our proposed parallel generation framework is not tied to MMaDA and can naturally extend to multimodal diffusion architectures that support continuous visual representations. For (3), recent emerging datasets and benchmarks such as [1] for multimodal mathematical reasoning offer promising opportunities. We plan to incorporate such data in future work to evaluate whether our parallel paradigm can also benefit math-related reasoning and generation tasks.
>
> [1]Li, Ang, et al. "Zebra-cot: A dataset for interleaved vision language reasoning." arXiv preprint arXiv:2507.16746 (2025).

---

> > ### Comment · Reviewer_WaTC · 2025-11-28
> >
> > Thank you to the authors for the hard work and new results, which have solved my concern. I will keep my positive rating.

---

> > > ### Author Response · Authors · 2025-11-28
> > >
> > > Dear Reviewer WaTC:
> > >
> > > We are very pleased that our rebuttal has addressed your concerns. Your feedback has been invaluable for our research!
> > >
> > > Warm regards,
> > >
> > > The Authors

---

### Official Review · Reviewer_rn82 · 2025-11-01

**Soundness:** 4
**Presentation:** 4
**Contribution:** 4
**Rating:** 6
**Confidence:** 3

**Summary:**

Based on the observation that existing AR models can suffer from error propagation during their thinking trajectory, this paper 1) proposes a benchmarking focusing on text image quality and output alignment, 2) develops a discrete-diffusion-based approach for parallel denoising of both text and images, and 3) introduces parallel RL to further optimize the intermediate cross-modal synergy.

**Strengths:**

The paper is well structured and clearly motivated. It offers a thorough investigation that includes new benchmarking, curated training datasets, new model design, and an RL protocol aimed at addressing the problem. The proposed MMaDA-Parallel model performs on par with SOTA open-source model that is trained on more data.

**Weaknesses:**

Normally, the decoding process only has one scheduler. In this paper, two schedulers are used for each modality. Could the authors give a more systematic guarantee of why we can assume the independence between each modality and why the alignment of text image modality would enable independent, parallel generation to work better than any-order joint generation?

**Questions:**

1. Please see Weaknesses.

2. In Table 3 (Table 6, 7 as well), why is MMaDA-Parallel decoding compared with the sequential decoding baseline that generates text before images? Why is it not compared to any-order joint decoding?

---

> ### Author Response · Authors · 2025-11-24
>
> We sincerely thank you for the thoughtful review and for recognizing the paper’s structure, motivation, new benchmarking, curated data, new model design, and RL protocol. We appreciate the positive assessment that MMaDA-Parallel performs on par with stronger open-source baselines trained on more data. Below we address your core concern.
>
> **Q: Questions and concerns about any-order joint generation**
>
> We thank the reviewer for this insightful suggestion. We explain this in both our design choices and additional experimental results.
> 1. **Design choices of why not use any-order joint generation**
>
> First, our core innovation is the exploration of the parallel generation paradigm in thinking-aware image editing and generation. To isolate this factor, we deliberately kept minimal changes to architectural and scheduler choices in our baseline, and directly followed the established modality-specific schedulers used in MMaDA.
>
> Second, there is a **mismatch in denoising steps across modalities**. Consistent with MMaDA, we empirically found that text denoising requires substantially longer steps than image denoising. For a sequence of length 1024, text requires more than 512 steps to stabilize semantics, whereas images reach high quality with far fewer steps (≈15–50). This makes designing a joint scheduler much harder, as it is required to
> **compromise** between these modalities. Any-order scheduling may lead to over-denoised images, which lowers efficiency, or to under-denoised text with much worse semantics.
>
> Third, for systematic guarantee, using separate schedulers **does not assume that the modalities are independent.**
> In unified discrete diffusion models such as MMaDA, the diffusion forward process is defined independently for each modality, since text tokens and visual tokens follow different noise distributions and degradation behaviors. This factorization in the forward process makes modality-specific schedules mathematically valid. Importantly, the reverse denoising steps **are not independent**: at every step, the text and image tokens interact through bidirectional cross-modal attention, sharing semantics throughout the trajectory. Therefore, the scheduler only controls **the pace of denoising** for each modality, while cross-modal attention ensures that the two modalities remain jointly aligned. This provides a systematic justification for modality-specific schedulers without assuming independence in generation.
>
>
> 2. **Additional experimental results of any-order generation**
>
> To empirically verify our hypothesis and address your concern, we trained a new baseline model using Any-Order Joint Generation (where text and image tokens are masked and predicted jointly with a single scheduler). We have added these experimental results in our revised paper **Page 26, Appendix I, first paragraph "Any-Order generation"**, both quantitatively and qualitatively in Figure 13 and Table 11. For convenience, we also provide the comparison below:
>
> **Table a: Comparison with Any Order Generation**
> | Schedulers             | Text Qual. | Text Align. | Image Cons. | Image Align. | Image Qual. | Output Align. | Overall |
> |------------------------|------------|-------------|-------------|--------------|-------------|----------------|---------|
> | Any-order   | 73.2       | 64.2        | 70.3        | 57.4         | 80.9        | 52.6           | 66.4    |
> | **Modal-Specific**     | **80.4**     | **71.0**    | **73.4**    | **63.2**       | **81.2**    | **59.8**       | **71.5** |
>
> These results show that the modality-specific scheduler achieves consistently better performance across all major dimensions, including reasoning quality, modality alignment, and overall output coherence. This provides strong empirical support for our design choice over any-order variants.

---

### Author Response · Authors · 2025-11-24
**Global Response**

We sincerely thank all the reviewers for the thorough reviews and valuable feedback. We are glad to hear that our findings and analysis are considered interesting and insightful (Reviewers WaTC, nFAx), the paper is well written and clearly motivated (Reviewers rn82, WaTC, nFAx), the visualizations are impressive(Reviewer WaTC), and that the paper is well written and well structured (Reviewers WaTC, w4zL). We are also grateful that reviewers recognize our work as introducing a new perspective for thinking-aware image synthesis with new benchmarking and a new generation paradigm (Reviewers rn82, w4zL). We have revised the manuscript according to the reviewers’ suggestions (**mainly in the appendix section, marked in blue**).

We here summarize and highlight our responses to the reviewers:
*  **We conducted additional ablations** on denoising schedulers, stronger LLMs for CoT labeling, and added more benchmarking results on general public benchmarks (Reviewers rn82, w4zL, WaTC).
*  **We provided further analysis** on failure cases, the generation process, and the training cost and complexity of our method (Reviewers WaTC, w4zL, nFAx).
* **We scaled our proposed method** to larger data, stronger visual tokenizers, and a more capable backbone, demonstrating clear scalability of our proposed parallel paradigm (Reviewers nFAx, w4zL).

We reply to each reviewer’s concerns in detail below their individual reviews.
Thank you again, and please feel free to reach out if you need any further clarification.

---

### Meta-Review · Area_Chair_9gV6 · 2026-01-07

**Summary:**

The AC carefully read the paper and the full discussion. The submission received mixed initial reviews (scores: 4, 6, 6, 6). Reviewers generally recognized the strong motivation and insightful analysis behind ParaBench, but raised several concerns: the use of two separate schedulers for different modalities, missing results on widely used benchmarks, limited novelty, and insufficient details on image quality, model naming, and the VLM models evaluated. In the rebuttal, the authors addressed the key issues raised during the initial review phase, and the updated, aggregated scores now trend toward acceptance. Accordingly, I am inclined to recommend acceptance

**Reviewer Concerns:**

Most of the concerns—about using two schedulers for different modalities (reviewer rn82), the absence of results on widely used benchmarks such as GenEval [1], WISE [2], and GEdit-Bench [3] (reviewer WaTC), the method’s novelty (noting it largely combines existing ideas in diffusion fine-tuning and reinforcement learning in a parallel setting) (reviewer nFAx), and missing details or clarifications about the VLM models as well as questions around image quality (reviewer w4zL)—have been addressed.

**Reviewer Scores:**

Reviewer rn82 is likely to maintain their positive assessment, as the authors clearly explained the “any-order joint generation” in the rebuttal. Reviewer WaTC is also expected to keep their favorable rating, given that the authors provided results on the requested benchmarks. Reviewer nFAx indicated they would raise their score. In contrast, reviewer w4zL suggested they would keep their original score unchanged.

---

### Decision · Program_Chairs · 2026-01-26

Accept (Poster)